# A quantitative model predicts how m6A reshapes the kinetic landscape of nucleic acid hybridization and conformational transitions

Bei Liu [1], Honglue Shi [2], Atul Rangadurai[1], Felix Nussbaumer[3], Chia-Chieh Chu[1,5], Kevin Andreas Erharter[3], David A. Case[4], Christoph Kreutz [3] & Hashim M. Al-Hashimi [1,2✉]

$N^6$-methyladenosine (m6A) is a post-transcriptional modification that controls gene expression by recruiting proteins to RNA sites. The modification also slows biochemical processes through mechanisms that are not understood. Using temperature-dependent (20°C–65°C) NMR relaxation dispersion, we show that m6A pairs with uridine with the methylamino group in the *anti* conformation to form a Watson-Crick base pair that transiently exchanges on the millisecond timescale with a singly hydrogen-bonded low-populated (1%) mismatch-like conformation in which the methylamino group is *syn*. This ability to rapidly interchange between Watson-Crick or mismatch-like forms, combined with different *syn:anti* isomer preferences when paired (~1:100) versus unpaired (~10:1), explains how m6A robustly slows duplex annealing without affecting melting at elevated temperatures via two pathways in which isomerization occurs before or after duplex annealing. Our model quantitatively predicts how m6A reshapes the kinetic landscape of nucleic acid hybridization and conformational transitions, and provides an explanation for why the modification robustly slows diverse cellular processes.

[1] Department of Biochemistry, Duke University School of Medicine, Durham, NC, USA. [2] Department of Chemistry, Duke University, Durham, NC, USA. [3] Institute of Organic Chemistry and Center for Molecular Biosciences Innsbruck (CMBI), University of Innsbruck, Innsbruck, Austria. [4] Department of Chemistry and Chemical Biology, Rutgers University, Piscataway, NJ, USA. [5] Present address: Department of Chemistry, University of Chicago, Chicago, IL, USA. ✉email: hashim.al.hashimi@duke.edu

N[6]-methyladenosine (m[6]A) (Fig. 1a) is an abundant RNA modification[1,2] that helps control gene expression in a variety of physiological processes including cellular differentiation, stress response, viral infection, and cancer progression[3–5]. m[6]A is also the most prevalent form of DNA methylation in prokaryotes where it is used to distinguish benign host DNA from potentially pathogenic nonhost DNA[6]. Although under debate[7], there is also evidence for m[6]A in mammalian DNA where it is proposed to play roles in transcription suppression and gene silencing[8,9].

In RNAs, m[6]A is thought to primarily function by recruiting proteins to specific modified sites (reviewed in refs. [3–5]). However, there is also growing evidence that the modification can impact a range of biochemical processes by changing the behavior of the methylated RNAs[10,11]. For example, by destabilizing canonical double-stranded RNA (dsRNA)[12], m[6]A has been shown to promote the binding of proteins to single-stranded regions of RNAs (ssRNAs)[10]. The modification has also been shown to slow biochemical processes that involve base pairing. For example, in messenger RNAs (mRNAs), m[6]A delays transfer RNA (tRNA) selection and reduces the translation efficiency in vitro[13] and in vivo[14] by 3–15-fold. In mRNA introns, m[6]A slows splicing and promotes alternative splicing in vivo[15]. Additionally, m[6]A reduces the rate of NTP incorporation during DNA replication[16] and reverse transcription[17] in vitro by 2–13-fold.

Recently, we developed and validated a nuclear magnetic resonance (NMR) relaxation–dispersion (RD)[18–20]-based method to measure the hybridization kinetics in DNA and RNA duplexes[21]. Using this approach, we showed that m[6]A preferentially slows the apparent rate of RNA duplex annealing by ~5–10-fold while having little effect on the apparent rate of duplex melting[21] (Fig. 1b). This impact of m[6]A on hybridization kinetics stands in contrast to mismatches that slow the rate of duplex annealing and also substantially increase the rate of duplex melting by up to ~100-fold[22–24]. How m[6]A selectively slows duplex annealing remains unknown. The comparable m[6]A-induced slowdown observed for duplex annealing and a variety of biochemical processes indicates that a common mechanism might be at play[13,16,17].

It has been known for many decades that the methylamino group of the m[6]A nucleobase can form two rotational isomers that interconvert on the millisecond timescale[25,26] (Fig. 1a). The preferred syn isomer[12,25,26] cannot form a canonical Watson–Crick base pair (bp) with uridine as the methyl group impedes one of the hydrogen bonds (H-bonds) (Fig. 1a). Rather, when paired with uridine, the methylamino group rotates into the energetically disfavored anti isomer and forms a canonical m[6]A–U Watson–Crick bp that retains both (A)N1···H-N3(U) and (A)N6···H-O4(U) H-bonds (Fig. 1a). As isomerization is energetically disfavored, it has been proposed to explain how m[6]A destabilizes dsRNA via the so-called "spring-loading"[12] mechanism despite forming a canonical Watson–Crick m[6]A–U bp.

Kinetic mechanisms involving binding and conformational change can occur via pathways wherein the conformational change occurs prior to or post binding[27]. We, therefore, hypothesized that m[6]A could slow hybridization via at least two pathways in which isomerization of the methylamino group occurs either before or following duplex formation (Fig. 1c). In the conformational selection (CS) pathway, hybridization proceeds via an unpaired intermediate (ssRNA[anti]) with m[6]A in the energetically disfavored anti conformation (Fig. 1c). In the induced-fit (IF) pathway, the more populated ssRNA[syn] species with m[6]A in the syn conformation initially hybridizes to form a double-stranded intermediate (dsRNA[syn]) that entails the loss of at least one Watson–Crick H-bond between m[6]A and the partner uridine (Fig. 1a). This is then followed by isomerization to form the Watson–Crick bp (dsRNA[anti]) with m[6]A in the anti conformation (Fig. 1c). To date, there has been no evidence for the dsRNA[syn] intermediate.

Here, using NMR RD, we show that m[6]A with the methylamino group in the anti conformation forms a Watson–Crick bp with uridine that transiently exchanges on the millisecond timescale with an unusual singly hydrogen-bonded, low-populated (1%), and mismatch-like conformation through isomerization of the methylamino group to the syn conformation. This ability to rapidly interchange between Watson–Crick or mismatch forms, combined with different syn:anti isomers preferences when paired versus unpaired, explains how m[6]A robustly and selectively slows

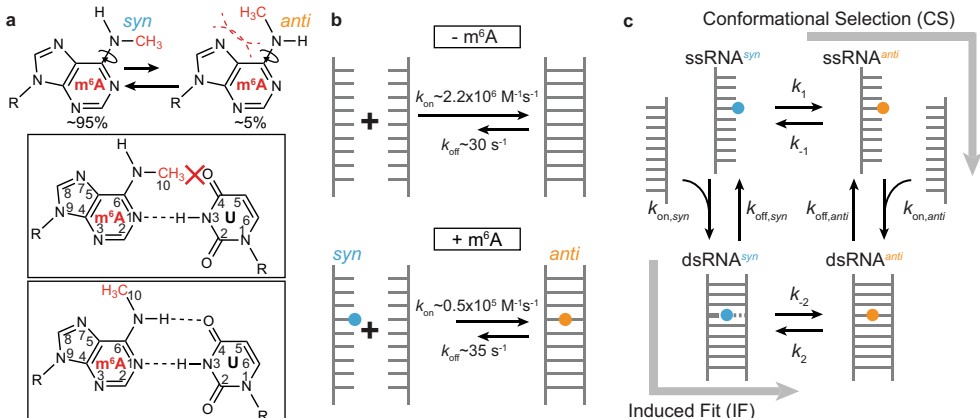

**Fig. 1 The syn and anti isomers of m6A. a** The m[6]A nucleobase shows a 20:1 preference for the syn isomer due to unfavorable steric interactions (shown in dashed red lines) in the anti isomer[12, 25]. In a duplex, the syn isomer impedes Watson–Crick pairing, and the anti isomer becomes the dominant form. **b** Apparent annealing ($k_{on}$) and melting ($k_{off}$) rate constants for unmethylated (−m[6]A) and methylated (+m[6]A) dsRNA. Rate constants shown were obtained from CEST measurements on dsGGACU with and without m[6]A at $T = 65$ °C[21]. **c** Schematic of the general four-state CS + IF model. $k_1$ and $k_{-1}$ are the forward and backward rate constants for methylamino isomerization in ssRNA, respectively; $k_2$ and $k_{-2}$ are the forward and backward rate constants for methylamino isomerization in dsRNA, respectively; $k_{on,anti}$ and $k_{off,anti}$ are the annealing and melting rate constants, respectively, when m[6]A adopts anti conformation in both ssRNA and dsRNA; $k_{on,syn}$ and $k_{off,syn}$ are the annealing and melting rate constants, respectively when m[6]A adopts syn conformation in both ssRNA and dsRNA.

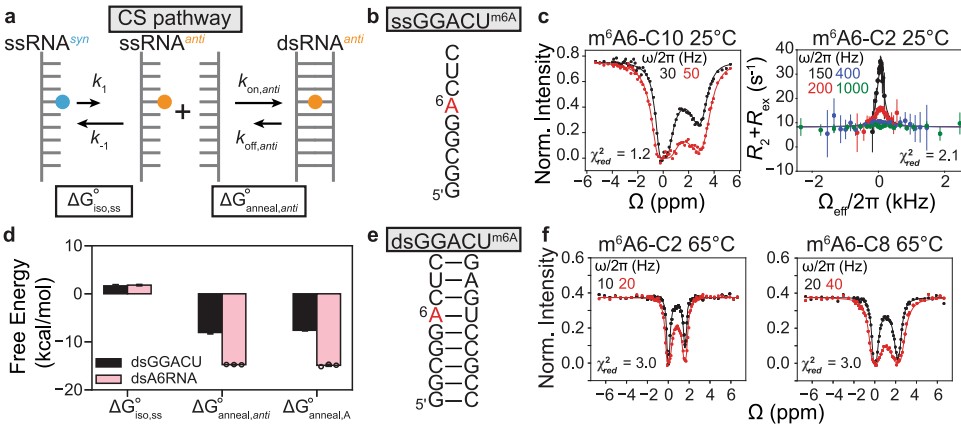

**Fig. 2 Testing a conformational selection kinetic model for m⁶A hybridization. a** The CS pathway. $\Delta G^{\circ}_{iso,ss}$ is the free energy of methylamino isomerization in ssRNA. $\Delta G^{\circ}_{anneal,anti}$ is the free energy of annealing the methylated ssRNA when m⁶A is *anti*. **b** ssGGACU sequence with the m⁶A site is highlighted in red. **c** ¹³C CEST profile for m⁶A6-C10 and off-resonance ¹³C $R_{1\rho}$ RD profile for m⁶A6-C2 in ssGGACU^m⁶A. **d** Free energy decomposition (see "Methods") of the CS pathway for dsGGACU^m⁶A at $T = 65\,^{\circ}C$ and dsA6RNA^m⁶A (Supplementary Fig. 1) at $T = 20\,^{\circ}C$. $\Delta G^{\circ}_{anneal,A}$ is the free energy of annealing unmethylated ssRNA and the value for dsGGACU was obtained from a prior study using RD measurements[21], and for dsA6RNA was measured using UV melting experiments (Supplementary Table 4). Data for $\Delta G^{\circ}_{iso,ss}$ were presented as mean values ± 1 s.d. from Monte Carlo simulations for one RD measurement. Data for $\Delta G^{\circ}_{anneal,A}$ were presented as mean values ± 1 s.d. from $n = 3$ independent UV measurements. The errors for $\Delta G^{\circ}_{anneal,anti}$ were propagated from $\Delta G^{\circ}_{iso,ss}$ and $\Delta G^{\circ}_{anneal,A}$. **e** The dsGGACU^m⁶A duplex with the m⁶A site highlighted in red. **f** ¹³C CEST profiles for m⁶A6-C2 and C8 in dsGGACU^m⁶A at $T = 65\,^{\circ}C$ (data obtained from a prior study[21]). Solid lines in panels (**c**, **f**) denote a two-state and constrained three-state fit to the CS pathway, using Bloch–McConnell equations as described in "Methods". Buffer conditions for NMR experiments are described in "Methods". RF field powers used for CEST and spin-lock powers used for $R_{1\rho}$ are color-coded. Data for CEST profiles (**c**, **f**) were presented as mean values ± 1 s.d. (smaller than data points) from $n = 3$ independent measurements of peak intensities at zero relaxation delay (see "Methods"). Data for the $R_{1\rho}$ profile in panel (**c**) were presented as mean values ± 1 s.d. from Monte Carlo simulations for one measurement as described in "Methods". Source data for panel (**d**) are provided in the Source Data file.

duplex annealing without affecting melting via two pathways in which isomerization occurs before or after duplex annealing. We develop a model that quantitatively predicts how m⁶A reshapes the kinetic landscape of nucleic acid hybridization, and that could explain why the modification robustly slows a variety of cellular processes. The model also predicts that m⁶A more substantially slows fast intramolecular RNA conformational transitions, and this prediction was verified experimentally by using NMR.

## Results

**Kinetics of m⁶A methylamino isomerization in ssRNA.** The ssRNA^anti which is the intermediate along the CS pathway has been extensively characterized in the past, whereas there is no evidence for the dsRNA^syn IF intermediate. We therefore initially examined whether the CS pathway alone could explain how and why m⁶A reduces the rate of duplex annealing while not affecting the melting rate. We developed a CS model which assumes that the minor *anti* isomer of m⁶A hybridizes with apparent annealing ($k_{on}$) and melting ($k_{off}$) rate constants similar to those of the unmethylated RNA. This assumption is reasonable given that like unmethylated adenine, the *anti* isomer forms a canonical m⁶A–U Watson–Crick bp when paired with uridine[11,12,25,26]. Since the *syn* isomer is incapable of Watson–Crick pairing with uridine, the model assumes that hybridization only proceeds via annealing of the single-strand containing the minor *anti* isomer (ssRNA^anti) through a CS-type pathway[27,28] (Fig. 2a). The apparent $k_{on}$ would then be reduced relative to the unmethylated RNA because the methylamino group has to rotate from the major *syn* to the minor *anti* isomer prior to hybridization (Fig. 2a). However, because *anti* is the preferred isomer in the canonical duplex, and because hybridization is rate-limiting under our experimental conditions (see below), the apparent $k_{off}$ would remain equivalent to that of the unmethylated duplex.

To test this CS model, we first used NMR RD to measure the isomerization kinetics in a ssRNA containing the most abundant m⁶A consensus sequence[1,2] in eukaryotic mRNAs (ssGGACU^m⁶A; Fig. 2b). This was important given that prior kinetic measurements of isomerization were performed on the m⁶A nucleobase dissolved in organic solvents and the kinetics may differ in ssRNA under aqueous conditions[25].

To enable the RD measurements, we used organic synthesis (see "Methods") to incorporate m⁶A ¹³C labeled at the base C2 and C8, or methyl C10 carbons (Supplementary Fig. 1) into ssGGACU. We then performed NMR chemical exchange saturation transfer (CEST)[29–31] and off-resonance spin relaxation in the rotating frame ($R_{1\rho}$) experiments[18–20] to measure the isomerization kinetics. NMR RD experiments can be used to characterize conformational exchange between a dominant ground-state (GS) and short-lived low-populated "excited-state" (ES). The $R_{1\rho}$ experiment measures the line-broadening contribution ($R_{ex}$) to the transverse relaxation rate ($R_2$) during a relaxation period in which a continuous radiofrequency (RF) field is applied with variable power ($\omega_{SL}$) and frequency ($\omega_{RF}$). The RF field reduces the $R_{ex}$ contribution in a manner dependent on $\omega_{SL}$ and $\omega_{RF}$ and the exchange parameters of interest (see below). The RD profiles are typically displayed by plotting the measured $R_2 + R_{ex}$ as a function of $\omega_{SL}$ and $\omega_{RF}$. For detectable exchange, a peak is observed centered at the difference between the chemical shift of the GS and ES ($-\Delta\omega$, assuming $\omega_{GS} = 0$ and $\omega_{ES} = \Delta\omega$). The CEST experiment measures the impact of conformational exchange on longitudinal GS magnetization during a relaxation period following application of a continuous RF field with variable power ($\omega_{SL}$) and frequency ($\omega_{RF}$). When applied on resonance with the ES, the RF field saturates the ES magnetization, and this saturation can be transferred via conformational exchange to the GS. This typically results in a reduced signal

intensity for the GS and a minor dip centered at $\omega_{ES} = \Delta\omega$ when the RF is on resonance with ES. A major dip is also observed centered at $\omega_{GS} = 0$ when the RF field is on resonance with the GS. The dependencies of $R_2 + R_{ex}$ ($R_{1\rho}$) or the GS signal intensity (CEST) on $\omega_{SL}$ and $\omega_{RF}$ can be fit to the Bloch–McConnell (B–M) equations[32] describing N-site exchange to determine exchange parameters of interest (see below). Together, $R_{1\rho}$ and CEST, which are optimized for different nuclei and exchange kinetics, allowed robust characterization of chemical exchange between the major GS *syn* methylamino and the low-populated and short-lived ES[33] *anti* methylamino isomer in unpaired m[6]A.

Shown in Fig. 2c on the left is the CEST profile recorded for the m[6]A-C10 methyl carbon in ssGGACU[m6A] as a function of RF. As is typical for CEST profiles, a major dip is observed when the RF field is on-resonance with the GS chemical shift at $\Delta\omega = 0$. In addition, a minor dip was observed indicative of conformational exchange with a sparsely populated ES. The dip was observed at a chemical shift $\Delta\omega_{C10} = \omega_{ES} - \omega_{GS} = 3$ p.p.m., which was in good agreement with the value predicted for the *anti* isomer ($\Delta\omega_{C10} = 3$–5 p.p.m.) using density functional theory (DFT) calculations[34] (see "Methods"). Shown in Fig. 2c on the right is the $R_{1\rho}$ profile measured for m[6]A-C2 in ssGGACU[m6A] as a function of RF field. A peak was observed at $-\Delta\omega_{C2} = 0.6$ p.p.m. indicative of conformational exchange. A similar C2 RD was observed in methylated but not unmethylated AMP, as expected if the RD is reporting on isomerization (Supplementary Fig. 2a).

Based on a two-state fit of the m[6]A-C10 and m[6]A-C2 RD data (Fig. 2c), the population of the ssRNA[anti] isomer in ssGGACU[m6A] was ~9% and the exchange rate for isomerization ($k_{ex} = k_1 + k_{-1}$, where $k_1$ and $k_{-1}$ are the forward and backward rate constants, respectively) was ~600 s[−1] at $T = 25$ °C (Supplementary Table 1). The population was ~2-fold higher than the value measured in the nucleobase in organic solvent (Fig. 1a)[25] while the exchange rate was ~20-fold faster, and in better agreement with values reported recently for ssDNA[35] (at $T = 45$ °C; Supplementary Table 1). Similar *syn*–*anti* isomerization kinetics were obtained for another different sequence (Supplementary Fig. 2b).

## m[6]A(*anti*)–U and A–U have similar thermodynamic stabilities in dsRNA

Before testing whether the CS model can predict the hybridization kinetics of methylated duplexes, we tested a thermodynamic prediction made by our model, namely that the energetics of annealing a single-strand containing the *anti* isomer of m[6]A should be similar to the energetics of annealing the unmethylated control. In this scenario, m[6]A destabilizes a duplex[12] primarily due to the conformational penalty ($\Delta G^\circ_{iso,ss}$) accompanying *syn* to *anti* isomerization in the ssRNA, which we have measured here for ssGGACU[m6A] using NMR RD.

To test this prediction, we decomposed (Fig. 2a) the overall annealing energetics ($\Delta G^\circ_{anneal,m6A} = -6.5 \pm 0.1$ kcal/mol) of methylated dsGGACU[m6A] (Fig. 2e) measured previously using melting experiments[21] into the sum of $\Delta G^\circ_{iso,ss} = 1.6 \pm 0.2$ kcal/mol plus the desired annealing energetics ($\Delta G^\circ_{anneal,anti}$) of m[6]A when it adopts the *anti* isomer,

$$\Delta G^\circ_{anneal,m6A} = \Delta G^\circ_{iso,ss} + \Delta G^\circ_{anneal,anti}$$

Indeed, we find that $\Delta G^\circ_{anneal,anti} = -8.1 \pm 0.2$ kcal/mol is similar to that measured for the unmethylated RNA $\Delta G^\circ_{anneal,A} = -7.6 \pm 0.1$ kcal/mol, with the methyl group being only slightly stabilizing within error by $0.5 \pm 0.2$ kcal/mol. A similar result was obtained for a different duplex (Fig. 2d) and a similar conclusion was also reached previously using the isomerization energetics measured in the nucleobase[25,26].

Therefore, with respect to the thermodynamics of annealing canonical duplexes, m[6]A in the *anti* isomer behaves similarly (within <0.5 kcal/mol) to unmethylated adenine and m[6]A primarily destabilizes dsRNA due to the conformational penalty accompanying isomerization, consistent with the previously proposed "spring-loading" mechanism[12]. Consistent with this interpretation, RD measurements on the m[6]A monomer reveal that 3 mM Mg[2+] stabilizes the *anti* relative to *syn* isomer by ~0.5 kcal/mol[36], and correspondingly, the destabilizing effects of m[6]A on RNA duplexes is reduced by ~0.2 kcal/mol in the presence of 3 mM Mg[2+] relative to the absence of Mg[2+] (Supplementary Table 1).

## Testing the CS kinetic model

Next, we tested whether the CS kinetic model could explain the impact of m[6]A on the hybridization kinetics of the dsGGACU[m6A] RNA measured recently using NMR RD[21]. These experiments were performed at $T = 65$ °C under conditions in which the duplex was the GS, and the ssRNA comprising two species in rapid equilibrium (ssRNA[syn] $\rightleftharpoons$ ssRNA[anti]) was the ES with a population of ~25%. The CEST experiments were performed at high temperatures because at 37 °C, the ssRNA is too lowly populated (<0.1%) and the hybridization is too slow (<50 s[−1]) to be effectively characterized by RD. Based on a two-state fit (dsRNA $\rightleftharpoons$ ssRNA) of the m[6]A6-C2 and m[6]A6-C8 RD data (Supplementary Fig. 3a), m[6]A reduced the apparent rate of dsGGACU[m6A] annealing ($k^{app}_{on,m6A}$) relative to the unmethylated control ($k_{on}$) by 5-fold while having little impact on the melting rate ($k^{app}_{off,m6A} \approx k_{off}$)[21].

We used the three-state CS model to simulate the m[6]A6-C8 and m[6]A6-C2 RD profiles measured for the methylated dsGGACU[m6A] duplex. The exchange parameters for the first isomerization step (ssRNA[syn] $\rightleftharpoons$ ssRNA[anti]) were fixed to the values determined independently from RD measurements on ssGGACU[m6A] (Supplementary Fig. 2c). $k_{off,anti}$ was assumed to be equal to $k_{off}$ measured for the unmethylated dsGGACU. This assumption is reasonable considering that hybridization is rate-limiting under our experimental conditions, and given the similarity between the experimentally measured $k_{off}$ for methylated and unmethylated duplexes[21]. The value of $k_{on,anti}$ was slightly adjusted relative to $k_{on}$ of the unmethylated control ($k_{on,anti} \approx 2 \times k_{on}$) to take into account small differences in their annealing energetics (Fig. 2a). The remaining NMR exchange parameters ($\Delta\omega$, $R_1$, and $R_2$ of GS and two ESs) for the hybridization and isomerization steps were fixed to the values obtained from the two-state fit of the RD data measured for dsGGACU[m6A] and ssGGACU[m6A] (see "Methods").

Interestingly, this simulation with no adjustable parameters satisfactorily reproduced the RD data with $\chi^2_{red} = 6.8$. This can be compared with $\chi^2_{red} = 3.3$ (Supplementary Fig. 3a) obtained from a two-state fit of the RD data with six adjustable parameters. As a negative control, the agreement deteriorated considerably ($\chi^2_{red} = 51.5$) (Supplementary Fig. 3b) when decreasing the exchange rate by 20-fold to mimic values observed for the nucleobase in organic solvents[25]. A constrained three-state fit to the RD data using the CS model in which the exchange parameters were allowed to vary within experimental error by 1 s.d., and in which the ratio (but not absolute magnitude) of $k_{on,anti}$ and $k_{off,anti}$ was constrained to preserve the free energy of the hybridization step improved the agreement to $\chi^2_{red} = 3.0$ (see "Methods" and Fig. 2f) and yielded $k_{on,anti} \approx 2 \times k_{on}$ and $k_{off,anti} \approx k_{off}$ (Supplementary Table 2). Therefore, even when it to comes to hybridization kinetics, m[6]A in the *anti* isomer behaves similarly to unmethylated adenine.

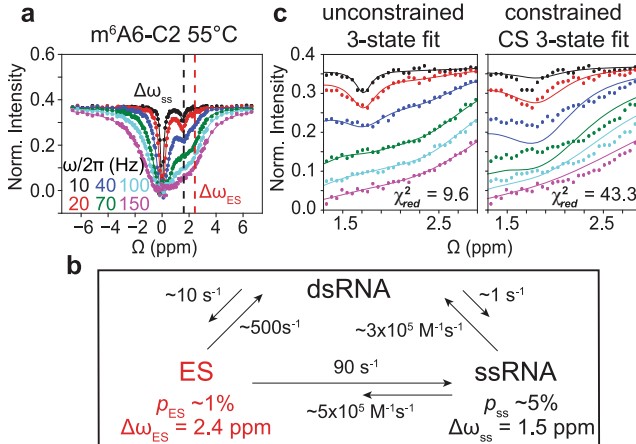

**Fig. 3 A hybridization intermediate for dsGGACU$^{m6A6}$ at $T = 55\,°C$. a** $^{13}C$ CEST profile for m6A6 C2 in dsGGACU$^{m6A6}$ at $T = 55\,°C$ shows a second dip at $\Delta\omega_{ES}$ that is distinct from the ssRNA ES at $\Delta\omega_{ss}$. **b** Exchange parameters (Supplementary Table 3) from a three-state fit to the RD data using a triangular model. **c** Zoom in to the m6A6 C2 CEST profiles comparing results from an unconstrained three-state fit to the Bloch–McConnell equations assuming the triangular model and a constrained three-state fit assuming a linear CS model. Data for CEST profiles were presented as mean values ± 1 s.d. (smaller than data points) from $n = 3$ independent measurements of peak intensities at zero relaxation delay (see "Methods").

These results provide a plausible explanation for the unique impact of m6A on RNA hybridization kinetics at $T = 65\,°C$. m6A does not impact the apparent melting rate because the dominant isomer in the duplex is *anti* and it melts at a rate comparable to that of the unmethylated RNA. On the other hand, m6A slows the apparent annealing rate by ~5-fold due to the ~10-fold lower equilibrium population of the ssRNA$^{anti}$ intermediate relative to the unmethylated ssRNA control and because the ssRNA$^{anti}$ intermediate anneals at a 2-fold faster rate relative to its unmethylated counterpart.

**An additional hybridization intermediate at $T = 55\,°C$.** Although we did not observe any evidence for the IF dsRNA$^{syn}$ intermediate, simulations indicate that its RD contribution was probably masked by the larger RD contribution from the ssRNA with a population ~22%. We therefore repeated the CEST measurements at a slightly lower temperature $T = 55\,°C$. This reduced the ssRNA population to ~5%, but it remained large enough to permit accurate measurements of hybridization kinetics using NMR RD. Repeating the measurements at a different temperature also allowed us to test the robustness of the CS model. Based on a two-state fit of the adenine C8 RD data, which only reports on a two-state hybridization process (Supplementary Fig. 4a), m6A reduced the apparent annealing rate by 20-fold while minimally (~1.6-fold) impacting the apparent melting rate under these conditions (Supplementary Fig. 4b).

Interestingly, we observed evidence for an additional ES, which manifested as a second minor dip in the m6A-C2 CEST profile (Fig. 3a). This ES dip at $\Delta\omega_{C2}$ ~2 p.p.m. was also observed at lower temperatures in another dsRNA (dsA6RNA$^{m6A}$) sequence context (Supplementary Fig. 5 and Supplementary Table 1). The fact that this ES was not observed in ssGGACU$^{m6A}$ indicated that it very likely was a dsRNA conformation. The ES was likely not observed at higher temperature $T = 65\,°C$ (Fig. 2f)[21] because it

was masked by the higher RD contribution from the more populated ssRNA ES.

The m6A-C2 RD data (Fig. 3a) could be satisfactorily fit to a three-state model that includes dsRNA, ssRNA, and the additional ES. Among several three-state topologies tested[37] (see Supplementary Fig. 4d), the best agreement was obtained with models that place the ES on-pathway between the dsRNA and ssRNA (Fig. 3b). Therefore, these results provide direct evidence for a dsRNA on-pathway hybridization intermediate and the CS pathway alone cannot fully explain the hybridization kinetics at $T = 55\,°C$. Indeed, simulations using the CS model did not reproduce the m6A-C2 RD data at $T = 55\,°C$ ($\chi^2_{red} \sim 600$) (Supplementary Fig. 4c) and neither did a constrained three-state fit to the CS model ($\chi^2_{red} \sim 43.3$) (Fig. 3c) because the model fails to account for the RD contribution from the additional ES.

**The dsRNA hybridization intermediate features an m6(*syn*) A⋯U stabilized by a single H-bond.** Understanding how m6A selectively slows annealing of dsGGACU at $T = 55\,°C$ by 20-fold without affecting the melting rate requires that we characterize the structure of the intermediate, which can be part of a separate hybridization pathway distinct from the CS pathway.

Although never observed previously, one possibility is that the intermediate is a dsRNA conformation in which the methylamino group rotates into the *syn* conformation. Such a conformation is predicted to be highly energetically disfavored in dsRNA, given the loss of at least one Watson–Crick H-bond. However, this loss in energetic stability would be partly compensated for by a gain in the stability of $\sim -1.5\,kcal/mol$ from restoring the energetically favored *syn* isomer. Such an intermediate would allow for an IF-type hybridization pathway, in which isomerization of the methylamino group occurs following and not before initial duplex formation (Fig. 1c).

To test this proposed conformation for the ES, we performed an array of NMR RD experiments using a stable hairpin variant of dsGGACU$^{m6A}$ (hpGGACU$^{m6A}$; Fig. 4a) with a much higher melting temperature ($T_m$ is predicted to be ~80 °C), designed to eliminate any background RD contribution from the ssRNA across a range of temperatures. Interestingly, we observed two-state RD for both m6A-C10 (Fig. 4b) and m6A-C2 (Supplementary Fig. 6a) at $T = 55\,°C$. A global fit of the data yielded an ES population (~1%), $k_{ex}$ (~500 s$^{-1}$), and $\Delta\omega_{C2} = 2.5$ p.p.m. that were in very good agreement with the values (Supplementary Table 3) measured for the on-pathway ES hybridization intermediate in dsGGACU$^{m6A}$. The $\Delta\omega_{C10}$ and $\Delta\omega_{C2}$ values were also in very good agreement with values predicted for m6(*syn*) A⋯U based on DFT calculations (Fig. 4g). Additional support that in the ES the methylamino group is *syn* comes from the kinetic rate constants of interconversion (Supplementary Note 1).

To gauge the nature of the Watson–Crick (m6A)N1⋯H3-N3(U) H-bond in the ES, we performed additional RD experiments targeting the N3 and H3 atoms of the partner uridine. We observed $^{15}N$ (Fig. 4c) and $^{1}H$ (Fig. 4d) RD only for the uridine partner of m6A (Supplementary Fig. 6a), and the two-state fit of the data yielded exchange parameters similar to those obtained from the carbon C2/C10 data (Supplementary Fig. 6a), indicating that they are reporting on the same ES. The $\Delta\omega_{N3} = -4.8$ p.p.m. and $\Delta\omega_{H3} = -3$ p.p.m. values indicated a substantial weakening of the remaining H-bond in the ES[38] (Fig. 4e). Indeed, a structural model for the m6(*syn*)A⋯U ES conformation that predicts the ES chemical shifts well based on DFT (Fig. 4g), and features a slightly (by 0.4 Å) elongated (m6A) N1⋯H3-N3(U) H-bond (Supplementary Fig. 6b). Note that while a minor peak was not observed in the $^{1}H$ CEST profile for U17-

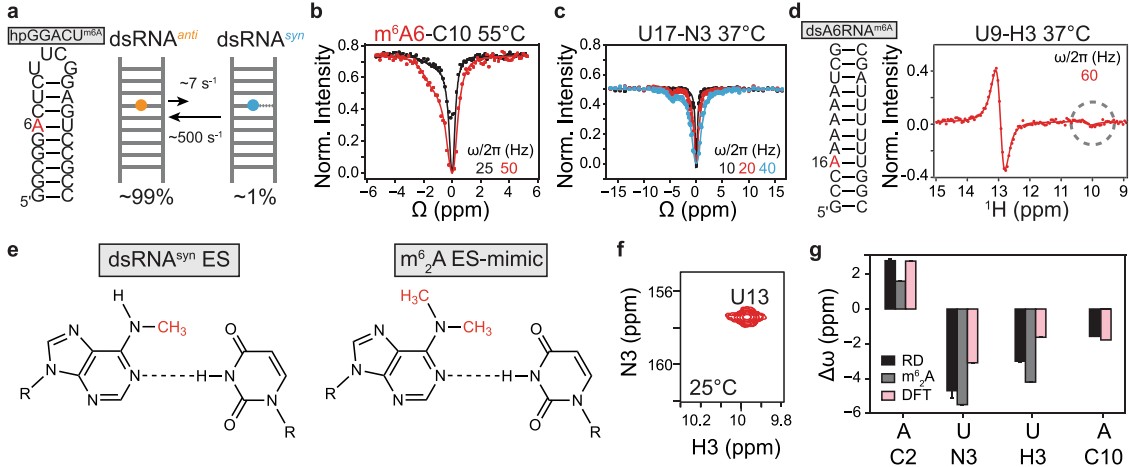

**Fig. 4 Characterizing the conformation of the ES intermediate. a** The hpGGACU$^{m6A}$ hairpin construct with the m6A site highlighted in red (left) and exchange parameters between dsRNA$^{anti}$ and dsRNA$^{syn}$ measured at $T = 55$ °C (right). **b** $^{13}$C CEST profile measured for m6A6-C10 in hpGGACU$^{m6A}$ at $T = 55$ °C. **c** $^{15}$N CEST profile measured for U17-N3 in hpGGACU$^{m6A}$ at $T = 37$ °C. **d** The dsA6RNA$^{m6A}$ duplex (left) and $^1$H CEST profile for U9-H3 at $T = 37$ °C (right). The minor peak is highlighted in the gray circle. **e** Chemical structures of proposed dsRNA$^{syn}$ ES and m$^6_2$A ES mimic. **f** 2D [$^{15}$N, $^1$H] HSQC spectra of U13-N3 $^{15}$N-site-labeled dsGGACU$^{m6_2A}$ at $T = 25$ °C. **g** Comparison of the chemical-shift differences ($\Delta\omega_{ES-GS} = \omega_{ES} - \omega_{GS}$) measured using RD in hpGGACU$^{m6A}$ (A C2/C10, U-N3) and dsA6$^{m6A}$ (U H3) at $T = 37$ °C (RD), when taking the difference between the chemical shifts measured for dsGGACU$^{m6_2A}$ and dsGGACU$^{m6A}$ (m$^6_2$A) and calculated using DFT as the difference between an m6(syn)A···U conformational ensemble and a Watson–Crick m6A(anti)–U bp (DFT) (see "Methods" section). Values for m$^6_2$A C10 are not shown because it is the site of modification. Solid lines in panels (**b–d**) denote a fit to the Bloch–McConnell equations to a two-state exchange model (see "Methods"). RF field powers for CEST profiles are color-coded. Data for CEST profiles in panels (**b–d**) were presented as mean values ± 1 s.d. (smaller than data points) from $n = 3$ independent measurements of peak intensities at zero relaxation delay (see "Methods"). Data for $\Delta\omega$ (panel **g**) were presented as mean values ± 1 s.d. from Monte Carlo simulations (number of iterations = 500) for one CEST measurement as described in "Methods" section. Source data for panel (**g**) are provided in the Source Data file.

H3 in hpGGACU$^{m6A}$, simulations indicate that this could be due to the 2-fold lower ES population (Supplementary Fig. 6c and Supplementary Table 1).

These results establish that the m6A methylamino group can also isomerize even in the context of a duplex m6(anti)A–U Watson–Crick bp and show that the preferences for the syn:anti isomers is inverted from ~10:1 in the unpaired single-strand to ~1:100 in the paired dsRNA.

**Chemical-shift fingerprinting the m6(syn)A···U ES using m$^6_2$A.**
To further verify the unusual m6(syn)A···U conformation proposed for the ES, we stabilized this species and rendered it the dominant conformation by replacing the m6A amino proton with a second methyl group so as to eliminate the GS Watson–Crick H-bond (Fig. 4e). This $N^6,N^6$-dimethyl adenine (m$^6_2$A) modification (Fig. 4e) is also a naturally occurring RNA modification[39].

Comparison of NMR spectra of dsGGACU with and without m$^6_2$A showed that the modification primarily affected the methylated bp while minimally impacting other neighboring bps (Supplementary Fig. 7a, c). Both the m$^6_2$A-C2 and U-N3 chemical shifts of the m$^6_2$A-modified dsGGACU (dsGGACU$^{m6_2A}$) were in very good agreement with those measured for the ES in dsGGACU$^{m6A}$ using RD (Fig. 4g). In addition, we observed an upfield shifted imino proton resonance (at ~10 p.p.m.), which could unambiguously be assigned via site labeling to the m6A partner U13-H3 (Fig. 4f and Supplementary Fig. 7a). This along with nuclear Overhauser effect-based distance connectivity (Supplementary Fig. 7a) indicate that the m6(syn)A···U ES likely retains a weaker (m6A6)N1···H-N3(U13) Watson–Crick H-bond, although we cannot rule out that the H-bond is mediated by water (see Supplementary Fig. 7e). Similar chemical-shift

agreement including for $\Delta\omega_{H3}$ was obtained for m$^6_2$A in dsA6RNA (Supplementary Fig. 7b).

Taken together, these data provide strong support for a singly H-bonded m6(syn)A···U bp (Fig. 4e), which is distinct from the bp open state (Supplementary Fig. 8 and Supplementary Note 2). To our knowledge, this alternative m6A-specific conformational state has not been documented previously.

**m6(syn)A···U behaves like a mismatch.** Although we initially dismissed hybridization pathways in which the major syn isomer hybridizes to form a dsRNA intermediate, our data indicate that this is indeed possible because m6A can pair with uridine to form the m6(syn)A···U conformation. Several lines of evidence indicate that m6(syn)A···U behaves like a mismatch when it comes to hybridization kinetics.

Like many mismatches[40], m6(syn)A···U loses a H-bond and is destabilized relative to the Watson–Crick m6(anti)A–U bp by ~3 kcal/mol. In addition, based on the three-state fit of the RD data measured for dsGGACU$^{m6A}$ at $T = 55$ °C (Fig. 3b), the m6(syn)A···U containing duplex intermediate anneals at a ~20-fold slower rate compared to the unmethylated control, whereas it melts with an ~80-fold faster rate. These changes in hybridization kinetics relative to the unmethylated control are also in line with those previously reported when introducing single mismatches to dsRNA[22–24].

We were able to verify the mismatch-like hybridization kinetics of m6(syn)A···U containing duplex by using NMR RD to measure the hybridization kinetics of the dsGGACU$^{m6_2A}$ ES mimic (Supplementary Fig. 7d). For dsGGACU$^{m6_2A}$, $k_{on}$ was ~16-fold slower, whereas $k_{off}$ was ~100-fold faster relative to the unmethylated RNA. Therefore, depending on the isomer, m6A can behave either like a Watson–Crick (anti) or mismatch (syn) when paired to the same partner uridine.

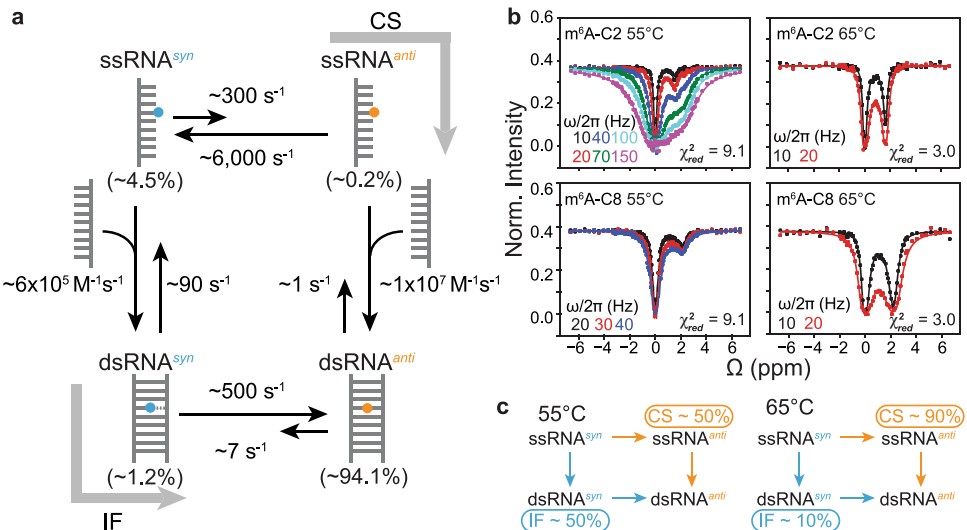

**Fig. 5 Testing a four-state CS + IF kinetic model. a** Schematic of the CS + IF model with populations and kinetic rate constants measured at $T = 55\,°C$ for dsGGACU$^{m6A}$. **b** Constrained four-state (CS + IF model) shared fit (solid lines) of the m6A C2 and C8 $^{13}$C CEST profiles to the Bloch–McConnell equations for dsGGACU$^{m6A}$ at $T = 55$ and $65\,°C$. $\chi^2_{red}$ values were obtained from global fitting m6A-C2 and m6A-C8 CEST data. RF field powers for CEST profiles are color-coded. Data for CEST profiles in panel (**b**) were presented as mean value ± 1 s.d. (smaller than data points) from $n = 3$ independent measurements of peak intensities at zero relaxation delay (see "Methods" section). **c** Equilibrium flux through CS and IF pathways at $T = 55$ and $65\,°C$.

**Kinetic model for m6A hybridization via conformation selection and IF**. The RD data measured for dsGGACU$^{m6A}$ at $T = 55\,°C$ provided direct evidence for hybridization via an IF pathway. However, the standalone IF pathway fails to account for the data measured at both $65\,°C$ (Supplementary Fig. 3c) and $55\,°C$ (Supplementary Fig. 4e) based on constrained fits. Since the RD data measured at $T = 65\,°C$ is consistent with hybridization via CS, with no evidence for flux along IF, we tested a general model that includes both pathways (CS + IF) (Fig. 5a).

We used the four-state CS + IF model along with the exchange parameters ($\Delta\omega$, $R_1$, and $R_2$ values) determined independently ("Methods") to simulate the RD data measured for dsGGACU$^{m6A}$ at $T = 55\,°C$. The exchange parameters associated with isomerization in ssRNA were again fixed to the values obtained from temperature-dependent RD measurements on ssGGACU$^{m6A}$ (Supplementary Fig. 2c). $k_{off,anti}$ was again assumed equal to $k_{off}$ and $k_{on,anti}$ was deduced by using the melting free energy obtained from RD measurements (see "Methods") (Fig. 2a). $k_{on,syn}$ and $k_{off,syn}$ describing the hybridization of ssRNA$^{syn}$ and methyl isomerization in dsRNA were fixed to the values obtained from the three-state fit of the RD data for dsGGACU$^{m6A}$ (Fig. 3b).

Indeed, the RD profiles simulated for m6A-C2 using the four-state model were in much better agreement ($\chi^2_{red} = 10.7$) (Supplementary Fig. 9a) with the experimental data relative to simulations using the CS model ($\chi^2_{red} = 563.7$) (Supplementary Fig. 4c) or constrained three-state fits to the CS model ($\chi^2_{red} = 43.3$) (Fig. 3c). A constrained fit of the RD data to the four-state model (see "Methods") improved the agreement further ($\chi^2_{red} = 9.6$) (Supplementary Fig. 9a) to a level comparable to the three-state fit (Fig. 3a). The $\chi^2_{red}$ values from globally fitting both m6A-C2 and m6A-C8 show similar trends (Fig. 5b).

These results provide a plausible explanation for how m6A selectively slows dsGGACU$^{m6A}$ annealing at $T = 55\,°C$ via both the CS and IF pathways. Based on optimized kinetic rate constants obtained from the constrained four-state fit of the RD data, the flux (see "Methods") was ~50:50 through the CS and IF

pathways at $T = 55\,°C$ (Fig. 5c). Along the CS pathway, m6A reduces the apparent rate of annealing due to the ~20-fold lower population of the ssRNA$^{anti}$ intermediate. However, as described for the data measured at $T = 65\,°C$, m6A does not affect melting because the dominant isomer in the duplex is anti, which behaves similarly to unmethylated adenine. Along the IF pathway, m6A reduces the apparent rate of annealing by 20-fold because m6($syn$)A···U behaves as a mismatch, reducing hybridization rate to form the dsRNA$^{syn}$ intermediate by 20-fold. Like a mismatch-containing duplex, this intermediate melts at a rate ~100-fold faster relative to the unmethylated duplex. However, the intermediate does not accelerate the apparent melting rate of the methylated duplex along the IF pathway relative to the unmethylated control because its equilibrium population is only ~1%.

We also reanalyzed the RD data measured at $T = 65\,°C$ and obtained good agreement with the constrained four-state fit ($\chi^2_{red} = 3.0$) (Fig. 5b). The level of agreement is similar to that obtained using the constrained three-state fit to the CS model (Fig. 2f), which is expected considering that majority (90%) of the flux is through the CS pathway (Fig. 5c). The smaller flux along the IF pathway at 65 versus $55\,°C$ can be attributed to a slower annealing rate along the IF pathway at $65\,°C$ due to a 2-fold reduction in the population of the ssRNA$^{syn}$ relative to ssRNA$^{anti}$ and comparatively 2.5-fold slower annealing rate constant of ssRNA$^{syn}$ along the IF pathway relative to ssRNA$^{anti}$ along the CS pathway.

**A quantitative model predicts how m6A reshapes the hybridization kinetics of DNA and RNA duplexes**. To test the generality and robustness of our proposed mechanism, we developed and tested a quantitative CS + IF model that predicts how methylating a central adenine residue impacts the hybridization kinetics for any duplex. The model assumes that the temperature-dependent isomerization kinetics in ssRNA and dsRNA does not vary, consistent with the small deviations (<2-fold) seen with sequence, as supported by our data (Supplementary Table 1). The model assumes that $k_{off,anti} = k_{off}$ and $k_{on,anti}$ is deduced based on

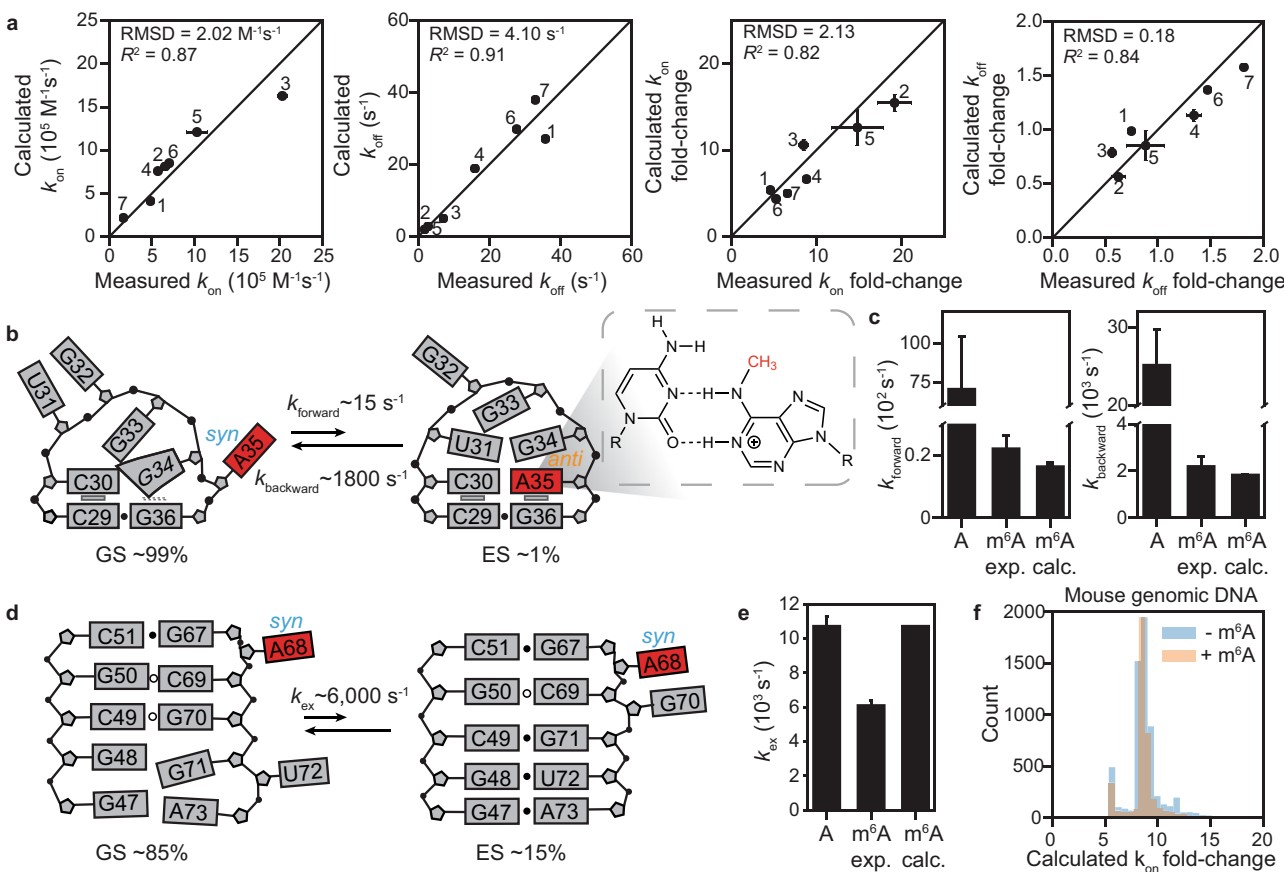

**Fig. 6 Testing the predictive power of the CS + IF model. a** Comparison of experimentally measured and predicted apparent $k_{on}$, $k_{off}$, and the fold-change relative to unmethylated duplex ($k_{on}$ fold-change = $k_{on}$(unmethylated)/ $k^{app}_{on,m^6A}$ and $k_{off}$ fold-change = $k_{off}$(unmethylated)\ $k^{app}_{off,m^6A}$) for RNA and DNA duplexes. Each point corresponds to a different duplex and/or experimental condition. All buffers contained 40 mM Na[+], unless stated otherwise: (1) dsGGACU$^{m^6A}$ at $T = 65$ °C, (2) at $T = 55$ °C, (3) with 3 mM Mg$^{2+}$ at $T = 65$ °C; (4) dsHCVm$^6$A with 3 mM Mg$^{2+}$ at $T = 60$ °C, (5) with 3 mM Mg$^{2+}$ at $T = 55$ °C, (6) with 3 mM Mg$^{2+}$ and 100 mM Na[+] at $T = 60$ °C; (7) dsA6DNA$^{m^6A}$ at $T = 50$ °C. Similar correlations were observed using RD simulation-based prediction method shown in Supplementary Fig. 9b. **b** Secondary structures of GS and ES in the apical loop of HIV-TAR with m$^6$A35 (highlighted in red), showing the chemical structure of the m$^6$A$^+$-C bp. **c** Comparison of $k_{forward}$ and $k_{backward}$ for unmethylated TAR (A), experimentally measured (m$^6$A exp.) and predicted (m$^6$A calc.) for methylated TAR. **d** Secondary structures of GS and ES of methylated RREIIB. **e** Comparison of $k_{ex}$ of unmethylated RRE (A), experimentally measured (m$^6$A exp.), and predicted (m$^6$A calc.) for methylated RRE. **f** Predicting the m$^6$A-induced slowdown effect on $k^{app}_{on,m^6A}$ of 12-mers (see "Methods") for m$^6$A sites[9] (orange) and random DNA (blue) in the mouse genome. Data in panels (**a**, **c**, **e**) were presented as mean values ±1 s.d. from Monte Carlo simulations (number of iterations = 500) for one CEST measurement as described in "Methods". Source data for panels (**a**, **c**, **e**) are provided in the Source Data file.

the known energetics of annealing the m$^6$A-containing duplex. The value of $k_{on,syn}$ was assumed to be 20-fold slower than the unmethylated RNA and $k_{off,syn}$ was then deduced by closing the thermodynamic cycle (see "Methods"). Using these rate constants and the CS + IF model, kinetic simulations (see "Methods") were used to predict $k^{app}_{on,m^6A}$ and $k^{app}_{off,m^6A}$.

We used the model to predict the $k^{app}_{on,m^6A}$ and $k^{app}_{off,m^6A}$ values recently reported[21] for two duplexes (dsGGACU$^{m^6A}$ and dsHCVm$^6$A) under a range of different salt (Mg$^{2+}$ and Na[+]) concentrations and temperatures and for an additional dataset involving dsHCVm$^6$A at $T = 55$ °C in 3 mM Mg$^{2+}$ (Supplementary Fig. 5). Across these duplexes and conditions, m$^6$A slowed the apparent annealing by ~5-fold to ~20-fold while minimally impacting the melting rate (<2-fold). As shown in Fig. 6a, a good correlation ($R^2 = 0.8$–0.9) was observed between the measured and predicted $k^{app}_{on,m^6A}$ and $k^{app}_{off,m^6A}$, as well as the overall impact on the apparent annealing and melting rates induced by methylation, with all deviations being <1.5-fold.

In all the above examples, the equilibrium flux was primarily (~50–95%) via the CS pathway. The differences in the m$^6$A-induced slowdown (~5–20-fold) of annealing across different duplexes are primarily driven by differences in the annealing rate of ssRNA$^{anti}$ along the CS pathway relative to that of unmethylated RNA, with the slowdown being more substantial the more stable the unmethylated duplex (Supplementary Fig. 9c). It should be noted that the slowdown is predicted to be even more substantial when hybridization is fast and isomerization of methylamino group becomes rate-limiting, as observed for an RNA conformational transition, as described below.

As an additional test, we used the model to predict the impact of m$^6$A on the apparent hybridization kinetics of an A-rich duplex DNA (dsA6DNA, Supplementary Fig. 5). Based on the unmethylated duplex hybridization kinetics measured previously[21], the model predicts that m$^6$A should reduce the apparent $k^{app}_{on,m^6A}$ by ~6-fold while having a little effect (<2-fold) on $k^{app}_{off,m^6A}$. We used NMR RD measurements (Supplementary

Fig. 5) on methylated dsA6DNA to test these predictions and the results show that m$^6$A reduces $k_\text{on,m$^6$A}^\text{app}$ by ~8-fold while having a little effect (<2-fold) on $k_\text{off,m$^6$A}^\text{app}$, in good agreement with the predictions (Fig. 6a).

Finally, we extended our model to also predict NMR CEST data by imposing additional constraints on NMR exchange parameters ($\Delta\omega$, $R_1$, and $R_2$) needed to simulate the RD data (see "Methods"). In addition to providing a rationale for the kinetic basis of the m$^6$A-induced hybridization slow down, such a model would also validate the existence of the IF and CS intermediates in diverse sequence contexts under a variety of experimental conditions. Thus, we subjected all of the above RD data to a constrained four-state fit to the CS + IF model. A reasonable fit ($\chi_\text{red}^2 \sim 3.5–14$) could be obtained in all cases (Supplementary Fig. 9d). This suggests that m$^6$A-induced hybridization slowdown in DNA is likely mediated by similar IF and CS intermediates as RNA.

**Testing kinetic model on RNA conformational transitions.** Beyond duplex hybridization, our kinetic model predicts that m$^6$A should also slow intramolecular conformational dynamics in which m$^6$A transitions between an unpaired conformation, in which the methylamino group is predominantly *syn*, and a paired conformation, in which the methylamino group is predominantly *anti*. In addition, the model predicts that the slowdown can be much more substantial for conformational transitions that are much faster than the hybridization kinetics measured under our experimental conditions.

To test these predictions, we methylated A35 in the apical loop of transactivation response element (TAR) (Fig. 6b) from human immunodeficiency virus type-1 (HIV-1)[41] and examined whether m$^6$A reduces the rate constant of a previously described intramolecular conformational transition in which unpaired A35 in the GS forms a wobble A35$^+$-C30 mismatch in the ES[42]. As in the Watson–Crick A–U bp, the methylamino group needs to be *anti* to form one of the H-bonds in the m$^6$A$^+$-C wobble (Fig. 6b). TAR therefore also allowed us to test the generality of the model to non-Watson–Crick bps.

We prepared a TAR NMR sample containing m$^6$A35 and $^{13}$C8-labeled G34 as an RD probe[42]. Based on the chemical-shift perturbations, m$^6$A destabilized the TAR ES relative to the GS by ~2 kcal/mol, in a manner analog to duplex destabilization[12] (Supplementary Fig. 10a and "Methods"). The CS + IF kinetic model predicts that m$^6$A will reduce $k_\text{ex}$, $k_\text{forward}$, and $k_\text{backward}$ for the TAR conformational transition by ~17-, ~400-, and ~14-fold, respectively. The much greater m$^6$A induced reduction in the forward rate constant relative to hybridization arises because the TAR conformational transition is intrinsically faster, and this pushes the isomerization step in the dominant CS pathway away from equilibrium, leading to a slowdown much greater than that due to the equilibrium population (~10%) of the ssRNA$^{anti}$ CS intermediate when hybridization is limiting. Here, the IF pathway is highly disfavored (flux < 1%) because the ES with m$^6$A in the *syn* conformation is predicted to be highly energetically disfavored.

Based on NMR RD measurements (Supplementary Fig. 10b), m$^6$A reduced $k_\text{ex}$, $k_\text{forward}$, and $k_\text{backward}$ by ~15-, ~300-, and ~12-fold in very good agreement with predictions from our model (Fig. 6c). The TAR experimental RD data could be satisfactorily fit to a constrained three-state fit to the CS model with $\chi_\text{red}^2 = 0.2$ (Supplementary Fig. 10c) comparable to that obtained from an unconstrained two-state fit. These results indicate that m$^6$A can also slow down RNA conformational transitions and potentially to a much greater degree than observed in our duplex hybridization experiments.

As a negative control, m$^6$A minimally (<2-fold) affects the exchange rate of conformational transition in the HIV-1 Rev response element stem IIB (RREIIB; Fig. 6d)[43] in which the m$^6$A remains unpaired in the two conformations (Fig. 6e, Supplementary Fig. 10d, and Supplementary Note 3).

**Discussion**
Our results help explain how m$^6$A selectively and robustly slows annealing while minimally impacting the rate of duplex melting under our experimental conditions. The minor ssRNA$^{anti}$ isomer hybridizes with kinetic rate constants similar to unmethylated adenine. m$^6$A slows the apparent annealing rate along the CS pathway relative to the unmethylated control due to the low equilibrium population of the ssRNA$^{anti}$ isomer. Once in a duplex, *anti* is the dominant isomer and m$^6$A does not substantially impact the apparent rate of duplex melting along the CS pathway. The major ssRNA$^{syn}$ isomer can also hybridize via an IF pathway to form a singly H-bonded bp and with kinetic rate constants similar to that of a mismatch-containing duplex. This intermediate forms slowly, explaining why m$^6$A also slows the apparent annealing rate along the IF pathway. However, because its equilibrium population is only ~1%, the intermediate does not accelerate the apparent melting rate along the IF pathway. While we have focused on relatively short duplexes with m$^6$A located in the middle, the impact of the modification on the hybridization kinetics will likely vary and be diminished when placed near the terminal ends, as observed for mismatches[22].

By treating the two m$^6$A isomers as two modular elements that have Watson–Crick or mismatch-like kinetic properties independent of sequence context[44], we were able to build a model that can predict the impact of m$^6$A on the overall hybridization kinetics and RNA conformational dynamics from component reactions. The power of such a quantitative and predictive kinetic model is that it obviates the need to carry out time-consuming kinetics experiments to measure the universe of kinetics data that is of biological interest. For example, when combined with an existing computational model that can predict the hybridization kinetics of unmethylated DNA duplexes from sequence[45], our model could be used to predict how a central m$^6$A impacts the hybridization kinetics of any arbitrary DNA duplex. This allowed us to predict the impact of m$^6$A on hybridization kinetics for all ~6000 m$^6$A sites reported in the mouse genome[9] (Fig. 6f). Our model may also aid the design and implementation of studies that harness the kinetic effects of m$^6$A as a chemical tool that can bring conformational transitions within detection or aid kinetic studies of RNA and DNA biochemical mechanisms.

Our model also makes a number of interesting biological predictions. The model predicts that m$^6$A should slow any process in which the unpaired m$^6$A in the predominantly *syn* isomer has to transition into a conformation in which m$^6$A is predominantly *anti*. This should include all templated processes that create canonical A–U Watson–Crick bps and many mismatches (A$^+$ (*anti*)–C (*anti*), A (*anti*)–G(*anti*), and A$^+$ (*anti*)–G (*syn*)), in which the methylamino group adopts the *anti* conformation. m$^6$A is found in a variety of RNAs involved in processes that require base pairing, including R-loop formation[46], microRNA RNA target recognition[47], snoRNA–pre-rRNA base pairing[48], snRNA–pre-mRNA base pairing[49], and the assembly of the spliceosome[50] and ribosome[51]. The model also predicts that the m$^6$A-induced slowdown could exceed 1000-fold for fast conformational transitions such as the folding of short hairpins and this could have important consequences on RNA folding, conformational switches, RNA protein recognition, and processes that occur co-transcriptionally. Further studies are needed to

examine whether m6A does indeed slow these processes and whether this has any biological consequences.

Our NMR measurements had to be performed under high-temperature conditions so that hybridization falls within the detection limits of RD. However, we were able to observe isomerization of the methylamino group in both ssRNA and dsRNA at $T = 37$ °C (Supplementary Figs. 2b and 6a). Based on the temperature dependence of the hybridization steps in the CS and IF pathways, our model predicts (see "Methods") that m6A will slow down annealing by ~5-fold while minimally impacting the melting rate consistent with our measurements at higher temperatures. A comparable level of the slowdown in annealing is also obtained when predicting the m6A-induced slowdown at $T = 37$ °C using rate constants for hybridization of unmethylated RNA reported previously[22] at $T = 37$ °C and assuming that m6A destabilizes dsRNA by 1 kcal/mol[52] (see "Methods").

The mismatch-like m6(syn)A···U bp is interesting not only because of its role in hybridization kinetics but also because it could potentially prime the methylamino group for recognition by reader proteins, which recognize the methylamino group in a syn conformation[53]. Upon surveying ~50,000 unmethylated A–U bps in Protein Data Bank (PDB), we found 428 bps that share the conformational signatures of the singly H-bonded m6A···U bp (see "Methods"). More than 60% of these bps are found in noncanonical regions, such as junctions, terminal ends, tertiary structural elements, and protein-bound RNA (Supplementary Fig. 7f). It will be interesting to examine whether the mismatch-like m6(syn)A···U forms as the dominant conformation in certain structural contexts where it may facilitate recognition by reader proteins both by locally destabilizing the bp, so that m6A is more accessible and by adopting a preformed syn conformation.

## Methods
### Sample preparation
*AMP and m6AMP.* Unlabeled adenosine and *N*6-methyladenosine 5′-monophosphate monohydrate (AMP and m6AMP) were purchased from Sigma-Aldrich (A2252 and M2780). Powders were directly dissolved in NMR buffer (25 mM sodium chloride, 15 mM sodium phosphate, 0.1 mM EDTA, and 10% $D_2O$ at pH 6.8 with or without 3 mM $Mg^{2+}$). The final concentrations of AMP and m6AMP were 50 mM.

*Oligonucleotides.* Unmethylated, methylated (*N*6-methylated adenosine, *N*6,*N*6-dimethyl adenosine), and 13C- or 15N-site-labeled (15N3-labeled U, 13C8,13C2-labeled A/m6A, and 13C10-labeled m6A) RNA oligonucleotides were synthesized using a MerMade 6 Oligo Synthesizer employing 2′-tBDSilyl-protected phosphoramidites and 1 μmol standard synthesis columns (1000 Å) (BioAutomation). Unlabeled m6A, m6₂A, rU, and *n*-acetyl-protected rC, rA, and rG phosphoramidites were purchased from Chemgenes. 15N3-labeled U and 13C8,13C2-labeled rA/m6A phosphoramidites were synthesized in-house according to published procedures[21,36]. 13C10-labeled m6A phosphoramidite was synthesized as described in Supplementary Note 4. RNA oligonucleotides were synthesized with the option to retain the final 5′-protecting group, 4,4′-dimethoxytrityl (DMT). Synthesized oligonucleotides were cleaved from columns using 1 ml AMA (1:1 ratio of 30% ammonium hydroxide and 30% methylamine), followed by 2-h incubation at room temperature. The solution was then air-dried and dissolved in 115 μl dimethyl sulfoxide, 60 μl triethylamine (TEA), and 75 μl TEA.3HF, followed by 2.5-h incubation at $T = 65$ °C for 2′-O deprotection. The solutions were then quenched using Glen-Pak RNA quenching buffer and loaded onto Glen-Pak RNA cartridges (Glen Research Corporation) for purification and subsequently ethanol precipitated. Following ethanol precipitation, RNA oligonucleotides were dissolved in water (200–500 μM for duplex samples, 50 μM for hairpin samples) and annealed by heating an equimolar amount of complementary single strands or hairpins at $T = 95$ °C for 10 min, followed by cooling at room temperature for 2 h for duplex samples or 30 min on ice for hairpin samples. Extinction coefficients for concentration calculation were obtained from the atdbio online calculator (https://www.atdbio.com/tools/oligo-calculator). The extinction coefficients for modified single strands were assumed to be equal to that of their unmodified counterparts. All samples were buffer exchanged using centrifugal concentrators (Amicon Ultra-15 3-kDa cut-off EMD Millipore) into NMR buffer (25 mM sodium chloride, 15 mM sodium phosphate, 0.1 mM EDTA, and 10% $D_2O$ at pH 6.8 with or without 3 mM $Mg^{2+}$).

The 13C8,13C2-labeled m6dA ssA6DNA oligonucleotide was synthesized in-house using a MerMade 6 oligo synthesizer. The 13C8,13C2-labeled m6dA phosphoramidite was synthesized as described in Supplementary Note 5. Standard DNA phosphoramidites (n-ibu-dG, bz-dA, ac-dC, and dT) were purchased from Chemgenes. DNA oligonucleotides were synthesized with the option to retain the final 5′-DMT group. Synthesized oligonucleotides were cleaved from columns using 1 ml AMA, followed by 2-h incubation at room temperature. The DNA sample was then purified using Glen-Pak DNA cartridges and ethanol precipitated. The complementary ssDNA of the m6A containing ssDNA is uniformly 13C/15N labeled and was synthesized and purified by in vitro primer (see Supplementary Table 8) extension[54] using 13C/15N isotopically labeled dNTPs (Silantes), and purified using 20% 29:1 polyacrylamide denaturing gel with 8 M urea, 20 mM Tris borate, and 1 mM EDTA, and electroelution (Whatmann, GE Healthcare) in 40 mM Tris acetate and 1 mM EDTA. DNA duplexes were prepared and buffer exchanged in a manner analogous to that described above for RNA duplexes.

### Definition of rate constants.

1. $k_1$ and $k_{-1}$ are the forward and backward rate constants for methylamino isomerization in ssRNA, respectively.
2. $k_2$ and $k_{-2}$ are the forward and backward rate constants for methylamino isomerization in dsRNA, respectively.
3. $k_{on}$ and $k_{off}$ are the annealing and melting rate constants, respectively, for unmethylated RNA.
4. $k_{on,anti}$ and $k_{off,anti}$ are the annealing and melting rate constants, respectively, when m6A adopts *anti* conformation in both ssRNA and dsRNA.
5. $k_{on,syn}$ and $k_{off,syn}$ are the annealing and melting rate constants, respectively, when m6A adopts *syn* conformation in both ssRNA and dsRNA.
6. $k_{on,m6A}^{app}$ and $k_{off,m6A}^{app}$ are the apparent annealing and melting rate constants, respectively, for m6A-methylated RNA.
7. $k_{forward}$ and $k_{backward}$ are the forward and backward rate constants, respectively, for conformational transitions measured using RD.

### NMR experiments
*Resonance assignments.* All NMR experiments (except for the imino proton exchange experiment) were performed on a Bruker Avance III 600 MHz spectrometer equipped with a 5 mm triple-resonance HCPN cryogenic probe. Resonance assignments for hpGGACUm6A have been reported previously[36]. Resonance assignments for m6₂A-modified dsGGACU and dsA6 were obtained using 2D [1H,1H] nuclear magnetic resonance spectroscopy experiments with 150 ms mixing time along with 2D [13C, 1H] and [15N, 1H] Heteronuclear single quantum coherence spectroscopy (HSQC) experiments. The assignments for ssGGACUm6A, ssA6RNAm6A, dsGGACU A/m6A, dsA6DNAm6A, and dsHCV A/m6A could be readily obtained since the samples were site-specifically labeled. The assignments for AMP and m6AMP were obtained from a prior study[25] (Supplementary Fig. 1). Data were collected using TopSpin 3.2 (Bruker), processed using NMRpipe software package[55], and analyzed using SPARKY (T.D. Goddard and D.G. Kneller, SPARKY 3, University of California, San Francisco).

*13C and 15N $R_{1\rho}$ relaxation dispersion.* 13C and 15N $R_{1\rho}$ experiments were performed using 1D $R_{1\rho}$ schemes as described previously[56–58]. The spin-lock powers ($\omega/2\pi$ Hz) and offsets ($\Omega_{eff}/2\pi$ Hz, where $\Omega_{eff} = \omega_{obs} - \omega_{rf}$, where $\omega_{obs}$ is the Larmor frequency of the spin and $\omega_{rf}$ is the carrier frequency of the applied spin-lock) are listed in Supplementary Table 5. The spin-lock was applied for a maximal duration (<120 ms for 15N and <60 ms for 13C) to achieve ~70% loss of peak intensity at the end of the relaxation delay.

*Analysis of $R_{1\rho}$ data.* 1D peak intensities were measured using NMRpipe[55]. $R_{1\rho}$ values for a given spin-lock power and offset combination were calculated by fitting the intensities as a function of delay time to a monoexponential decay[34]. A Monte Carlo approach was used to calculate $R_{1\rho}$ uncertainties[59]. Alignment of initial magnetization during the B–M fitting was performed based on the $k_{ex}/|\Delta\omega_{major}|$ ratio ($k_{ex}/|\Delta\omega_{major}| \geq 1$ and $k_{ex}/|\Delta\omega_{major}| > 1$ corresponding to GS alignment and AVG alignments, respectively)[18]. Chemical exchange parameters were obtained by fitting experimental $R_{1\rho}$ values to numerical solutions of the B–M equations[32] that describe N-site chemical exchange[34]. Errors in exchange parameters were determined using a Monte Carlo approach[34]. When available, $R_{1\rho}$ data measured for the same exchange process under the same condition were globally fitted, sharing ES population and exchange rate constants. Reduced $\chi^2$ ($\chi_{red}^2 = \sum_{i=1}^{N} \left( \frac{R_{1\rho(i)}^{Calc} - R_{1\rho(i)}^{exp}}{\sigma_{exp(i)}} \right)^2$, $R_{1\rho(i)}^{Calc}$ and $R_{1\rho(i)}^{exp}$ are experimentally measured and calculated $R_{1\rho}$ data using the B–M equations, $\sigma_{exp(i)}$ is the experimental uncertainty in $R_{1\rho}$ determined using a Monte Carlo approach) was calculated to assess the goodness of fitting[18]. In

general, similar exchange parameters were obtained from individual fitting and global fitting. All exchange parameters are summarized in Supplementary Table 1.

*Estimate p_ES of methylated TAR from chemical shifts.* The RD signal of methylated TAR is weak probably due to small $p_{ES}$ and fast $k_{ex}$ (Supplementary Fig. 10b). We used chemical-shift perturbation-based method[60] as an alternative approach to estimate the population of ES[42] ($p_{ES,m^6A}$) of methylated TAR. Specifically, in methylated TAR, $\omega_{obs} = \omega_{GS} \times (1 - p_{ES,m^6A}) + \omega_{ES} \times p_{p_{ES,m^6A}}$. $\omega_{GS}$ and $\omega_{ES}$ are chemical shifts of GS and ES of unmethylated TAR and were determined previously[60]. Based on 2D [13C, 1H] HSQC spectra, G34-C8 peak shifts toward GS (Supplementary Fig. 10a), and the calculated $p_{ES,m^6A}$ is ~1%.

*13C and 15N CEST.* 13C and 15N CEST experiments were performed using 1D schemes without equilibration of GS and ES magnetization prior to the relaxation delay[21]. The RF field strengths ($\omega/2\pi$ Hz) and offset combinations ($\Omega/2\pi$ Hz, where $\Omega = \omega_{rf} - \omega_{obs}$) used in CEST measurements are listed in Supplementary Table 6. The relaxation delay for all CEST experiments was 200 ms.

*2D CEST for 13C methyl probes.* The pulse sequence for the 13C methyl CEST was derived by modifying the 2D CEST experiment for 13C from Zhang and co-workers[29] in accordance with considerations described in Bouvignies and Kay[31] outlining a 2D CEST experiment for 13C methyl groups. The following changes were made to the CEST experiment from Zhang and co-workers[29]:

- Given that the samples for methyl CEST in this study were site specifically 13C labeled at the methyl group, we removed shaped pulse c that was used to refocus carbon–carbon scalar couplings.
- The delay $\tau$ between 13C pulses of phases φ2 and φ3 and φ3 and φ5 was set to be as close as possible to the optimal value of $\tau = \frac{arccos\sqrt{2/3}}{2\pi J_{HC}}$, where $J_{HC}$ is the scalar coupling between the methyl carbon and protons, and for optimal transfer of in-phase methyl carbon magnetization to antiphase, as described by Bouvignies and Kay, while ensuring that the delays between the pulses in the sequence were positive. $J_{HC}$ was measured using an F1-coupled 2D [13C, 1H] HSQC experiment.
- The $\tau$ delay flanking shaped pulse b was set to be equal to $\frac{arccos\sqrt{1/3}}{2\pi J_{HC}}$. The duration of shaped pulse b was shortened as needed so as to ensure that the delays between the pulses in the sequence were positive.
- A gradient pulse was inserted between the 13C and 1H $\pi/2$ pulses after T1 evolution, as described by Bouvignies and Kay[31], to purge transverse magnetization.

*Analysis of the CEST data.* 1D or 2D peak intensities were calculated using NMRpipe[55]. The intensity error for all offsets for a given spin-lock power was set to be equal to the standard deviation of three measurements of peak intensity with zero relaxation delay under the same spin-lock power. The intensities were normalized to the average intensity of the three zero delay measurements. Exchange parameters were then obtained by fitting experimental intensity values to numerical solutions of the B–M equations and RF field inhomogeneity was taken into account during CEST fitting[61]. No equilibration of GS magnetization was assumed when integrating the B–M equations for non-methyl probes[61], while equilibration was assumed for the methyl CEST given that the sequence employs nonselective hard pulses. Fits of CEST data were carried out assuming unequal $R_2$ or assuming equal $R_2$ for duplex melting[21] and other ES measurements, respectively. Alignment of the initial magnetization during CEST fitting was chosen based on the $k_{ex}/|\Delta\omega_{major}|$ ratio as described in the previous section[61]. Errors in exchange parameters were determined using a Monte Carlo approach with 500 iterations[62]. Global fitting of CEST data was carried out for the same exchange process under identical conditions. $\chi^2_{red}$ was calculated to assess the goodness of fitting as described in the previous section[18]. Note that the different $\chi^2_{red}$ values for different fits are most likely due to differences in the quality of the NMR data and poor estimation of the real experimental uncertainty (Supplementary Table 1). Model selection (three state with triangular, linear, or starlike topology; Supplementary Fig. 4d) was carried out by calculating Watanabe–Akaike information criterion and Watanabe–Bayesian information criterion weights for each model and selecting the model with the highest relative probability[34].

*1H CEST experiment.* A transverse relaxation optimized spectroscopy-based spin-state selective 1H CEST experiment[63] was carried as described previously[64]. The power of the $B_1$ field was set to be 60 or 120 Hz and the offset of the $B_1$ field ranged from 8.5 p.m. to 15.5 p.p.m. with a step of 30 Hz. The relaxation delay was 400 ms. The 1H CEST data were collected in a pseudo-3D mode and were analyzed using NMRpipe[55]. The intensities in the Nα and Nβ CEST profiles were normalized to a reference intensity with $B_1$ frequency $= -20$ p.p.m. The Nβ CEST profile was then subtracted from the Nα CEST profile to result in a difference CEST profile, from which the $\Delta\omega$ of the ES was fitted with predetermined fitting parameters such as $p_{ES}$, $k_{ex}$, and 15N $R_1$ from the 13C/15N $R_{1\rho}$ experiments. Errors in the CEST intensity profiles were estimated based on the scatter in regions of 1D profiles that

did not contain any intensity dips. The Python package *ChemEx* (https://github.com/gbouvignies/chemex) is used to carry out the fitting.

*Imino proton exchange experiment.* Experiments were carried out on a 700 MHz Bruker NMR spectrometer equipped with hydrogen cyanide room-temperature probe to measure the proton exchange between imino proton and water[65], following the same pulse programs and protocols as described in a prior study[66]. Briefly, the water proton longitudinal relaxation rate constant $R_1$ was first measured using a standard saturation-recovery method[66]. A pre-saturation pulse was used for solvent suppression. The relaxation delay time for measuring water proton $R_1$ was set to be 0.0, 0.4, 0.8, 1.2, 1.6, 2.0, 2.4, 2.8, 3.2, 3.6, 4.0, 4.4, 4.8, 5.2, 6.0, 7.0, 8.0, 9.0, 10.0, 12.0 and 15.0 s. The apparent solvent exchange rate constant of the imino protons was then measured using an inversion-recovery scheme by initially selectively inverting the bulk water magnetization, followed by detecting transfer of the water magnetization to the imino proton during the solvent exchange. A sinc-shaped $\pi$-pulse was optimized and used to invert the water magnetization. A binomial water-suppression scheme was used to suppress water. The delay times used to measure water and imino proton exchange rate constants are listed in Supplementary Table 7.

The apparent exchange rate ($k_{ex}$) of imino and water proton was obtained by fitting the imino magnetization as a function of exchange time upon solvent exchange according to Eq. (1),

$$W(t) = W^0 - E \times W^0 \times \frac{k_{ex}}{R_{1w} - R_{1n}} \times (e^{-R_{1n} \times t} - e^{-R_{1w} \times t}) \quad (1)$$

where $W(t)$ is the imino peak volume as a function of exchange time $t$, $W^0$ is the initial peak volume (at $t = 0$ s), $E$ is the efficiency of the inversion pulse, $k_{ex}$ is the apparent solvent exchange rate constant between imino and water proton, $R_{1w}$ is water proton $R_1$, $R_{1n}$ is the summation of imino proton $R_1$, and exchange rate constant $k_{ex}$. In the equation, $R_{1w}$ and $E$ values are fixed parameters that are predetermined, while $k_{ex}$ and $R_{1n}$ are fitted parameters. The error of the fitted parameters is the standard fitting error, which is the square root of the diagonal elements of the covariance matrix. The efficiency of the selective shaped pulse used for water inversion ($E$) was calculated by Eq. (2):

$$E = 1 - \frac{W_{inv}}{W_{eq}} \quad (2)$$

where $W_{inv}$ and $W_{eq}$ represent the peak volumes of the water proton with and without the shaped pulse for inversion, respectively (at zero delay time and without binomial water suppression).

Determining the methylamino isomerization rate constants from temperature-dependent RD measurements for methylated ssRNA and dsRNA. The observed temperature dependence of $k_1$, $k_{-1}$ in m6AMP and ssRNA (Supplementary Fig. 2c) and $k_2$, $k_{-2}$ in dsRNA (Supplementary Fig. 6d) determined using RD were fit to a modified van't Hoff equation that accounts for statistical compensation effects and assumes a smooth energy surface[57]:

$$ln\left(\frac{k_i(T)}{T}\right) = ln\left(\frac{k_B\kappa}{h}\right) - \frac{\Delta G_i^{\circ\,T}(T_{hm})}{RT_{hm}} - \frac{\Delta H_i^{\circ\,T}}{R}\left(\frac{1}{T} - \frac{1}{T_{hm}}\right) \quad (3)$$

where $k_i$ ($i = 1, -1$ or $2, -2$) is the rate constant, $\Delta G_i^{\circ\,T}$ and $\Delta H_i^{\circ\,T}$ are the free energy and enthalpy of activation ($i = 1, 2$) or deactivation ($i = -1, -2$), respectively, $R$ is the universal gas constant (kcal/mol/K), $T$ is temperature (K), and $T_{hm}$ is the harmonic mean of the experimental temperatures ($T_i$ in K) computed as $T_{hm} = n/\sum_{i=1}^{n}(1/T_i)$, $k_B$ is the Boltzmann's constant, $\kappa$ is the transmission coefficient (assumed to be 1). The goodness-of-fit indicator $R^2$ between the measured and fitted rate constants was calculated as follows: $R^2 = 1 - \frac{SS_{res}}{SS_{total}}$, $SS_{res} = \sum(k_{i,fit} - k_{i,exp})^2$, $SS_{total} = \sum(k_{i,exp} - \overline{k_{i,exp}})^2 \cdot k_{i,fit}$, and $k_{i,exp}$ ($i = 1, -1$ or $2, -2$) are fitted and experimentally measured rate constants. $\overline{k_{i,exp}}$ is the mean of all $k_{i,exp}$. Errors of fitting for $\Delta G_i^{\circ\,T}$ and $\Delta H_i^T$ were calculated as the square root of the diagonal elements of the covariance matrix. Given these fitted $\Delta G_i^{\circ\,T}$ and $\Delta H_i^T$ values, $k_i$ at $T = 55$ and 65 °C used for kinetic modeling was computed using Eq. (3).

*Measuring the kinetics of duplex hybridization from CEST data.* $k_{off}$ (s⁻¹) and $k_{on}$ (M⁻¹ s⁻¹) for duplex hybridization were determined based on the forward rate ($k_{forward}$) and backward ($k_{backward}$) rate constants obtained from a two-state fit of the dsHCV/dsHCVm6A A11-C8 and dsA6DNA m6A16-C2 RD data (two-state fit of other constructs were reported previously[21]) and a three-state fit of m6A-C2 dsGGACU$^{m^6A}$ at $T = 55$ °C:

$$k_{forward} = k_{off} \quad (4)$$

$$k_{backward} = k_{on} \times [ss2] \quad (5)$$

where [ss2] is the free concentration of the complementary single strand.

$$k_{backward} = k_{ex}(1 - p_{ss}) \quad (6)$$

where $p_{ss}$ is the single-strand population. The annealing rate constant $k_{on}$ is given by:

$$k_{on} = \frac{k_{ex}(1 - p_{ss})}{[ss2]} \quad (7)$$

The uncertainty in [ss2], and $p_{ss}$ and $k_{ex}$ from CEST measurements were propagated to determine the uncertainty in of $k_{on}$. From two-state CEST fit, $[ss2] = C_t \times p_{ss}$, where $C_t$ is the total concentration of the duplex, which was obtained using the extinction coefficient as described in the "Sample preparation" section. The uncertainty of $C_t$ was assumed to be 20%[21]. [ss2] from a three-state fit were calculated as described in the energetic decomposition section below.

*Validation of NMR RD measurements on m⁶A RNA hybridization.* We have previously[21] shown that hybridization kinetics measured from NMR RD on unmodified DNA and RNA duplexes are consistent with those measured using other techniques employing fluorescence spectroscopy. As an additional test, we performed temperature-dependent RD measurements for dsGGACU^{m⁶A} (Supplementary Fig. 11a). The annealing rate constant $k_{on}$ did not have a strong temperature dependence, consistent with prior studies reporting non-Arrhenius behavior for $k_{on}$ in unmodified duplexes[67,68]. On the other hand, the melting rate constant $k_{off}$ showed a strong temperature dependence, which was also consistent with prior studies[67,68]. The extrapolated annealing thermodynamic parameters including $\Delta G^\circ_{anneal}$, $\Delta H^\circ_{anneal}$, and $\Delta S^\circ_{anneal}$ measured from NMR experiments are in good agreement with those measured from ultraviolet (UV) melting experiments[36] (Supplementary Fig. 11b, c). We also observed a good agreement between the annealing free energy ($\Delta G^\circ_{anneal}$) measured using CEST and UV melting experiments for nine additional DNA/RNA duplexes at temperatures ranging from 45 to 65 °C[21] (Supplementary Fig. 11d).

**UV melting experiments.** UV melting experiments were conducted on a Perkin-nElmer Lambda 25 UV/VIS spectrometer with an RTP 6 Peltier Temperature Programmer and a PCB 1500 Water Peltier System. At least three measurements were carried out for each sample (3 μM in NMR buffer without $D_2O$) with a volume of 400 μl in a Teflon-stoppered 1 cm path length quartz cell. The absorbance at 260 nm ($A_{260}$) was monitored at temperatures ranging from 15 to 95 °C, and at a ramp rate of 1.0 °C/min. The melting temperature ($T_m$) and standard enthalpy change ($\Delta H^\circ$) of hybridization reaction for duplexes were obtained by fitting the absorbance of the optical melting experiment to Eqs. (8) and (9)[69],

$$A_{260} = ((m_{ss} \times T) + b_{ss}) \times p_{ss} + ((m_{ds} \times T) + b_{ds}) \times (1 - p_{ss}) \quad (8)$$

$$p_{ss} = 1 - \frac{1 + 4e^{\left(\frac{1}{T_m} - \frac{1}{T}\right)\frac{\Delta H^\circ}{R}} - \sqrt{1 + 8e^{\left(\frac{1}{T_m} - \frac{1}{T}\right)\frac{\Delta H^\circ}{R}}}}{4e^{\left(\frac{1}{T_m} - \frac{1}{T}\right)\frac{\Delta H^\circ}{R}}} \quad (9)$$

where $m_{ss}$, $b_{ss}$, $m_{ds}$, and $b_{ds}$ are coefficients describing the temperature dependence of the molar extinction coefficient of single strands and double strands, respectively, $T$ is the temperature (K), $R$ is the gas constant (kcal/mol/K), and $p_{ss}$ is the population of the single strand. Standard entropy change ($\Delta S^\circ$) and $\Delta G^\circ$ of double-strand hybridization were therefore computed from Eqs. (10) and (11):

$$\Delta S^\circ = \frac{\Delta H^\circ}{T_m} - R\ln\left(\frac{C_t}{2}\right) \quad (10)$$

$$\Delta G^\circ = \Delta H^\circ - T\Delta S^\circ \quad (11)$$

where $C_t$ is the total concentration of duplex. The uncertainty in $T_m$ and $\Delta H^\circ$ were obtained based on standard deviation in triplicate measurements that were propagated to the uncertainty of $\Delta S^\circ$ and $\Delta G^\circ$.

**MD simulations.** To generate an ensemble of RNA duplexes with different m⁶A geometries, we performed MD simulations on dsGGACU with the m⁶A–U bp in either *syn* or *anti* conformations, or an m⁶₂A···U bp. All MD simulations were performed using the ff99 AMBER force field with bsc0 and χ_{OL3} corrections for RNA, using periodic boundary conditions as implemented in the AMBER MD package. Starting structures for MD of unmethylated dsGGACU were generated by building an idealized A-RNA duplex using the *fiber* module of the 3DNA suite of programs[70]. The starting structures for dsGGACU^{m⁶A} with an m⁶A–U bp in either the *anti* or *syn* conformation were generated by replacing the *anti* and *syn* adenine amino hydrogen atoms in the idealized unmethylated dsGGACU structure with a methyl group. The starting structure for the dsGGACU duplex with the m⁶₂A–U bp was generated by replacing both of the amino hydrogen atoms of the adenine in the idealized unmethylated dsGGACU structure with methyl groups. All starting structures were solvated with an octahedral box of SPC/E water molecules with box size chosen such that the boundary was at least 10 Å away from any of the DNA atoms. Na⁺ ions treated using the Joung–Cheatham parameters were then added to neutralize the charge of the system. The system was then energy minimized in two stages with the solute heavy atoms (except for the atoms comprising the m⁶₂A···U bp and the m⁶(*syn*)A···U bp being fixed (with a restraint of 500 kcal/mol/Å²)

during the first stage. Heating, equilibration, and production runs (500 ns) were performed as described previously[71]. To maintain the methyl group in the *syn* conformation during the MD simulation of the dsGGACU duplex with the m⁶(*syn*) A···U bp, a torsion angle restraint was applied on the angle spanning the methyl carbon-N6-C6-C5 atoms of m⁶A. The restraint was chosen to be square welled between 160° and 200°, parabolic between 159–160° and 200–201°, and linear beyond 201° and <159°, with a force constant of 32 kcal/mol/Å². Force field parameters for m⁶A were derived from those by Aduri et al.[72]. In particular, the atom types and charges for the methyl group were taken from those by Aduri et al., while retaining atom types and charges (apart from N6, see below) for the remaining atoms from those of adenine in the AMBER ff99bsc0χOL3 force field. Charges on the amino N6 atom of m⁶A were adjusted to maintain a net charge for the m⁶A nucleoside of −1. An analogous procedure was followed to generate the parameters for the m⁶₂A nucleoside. Missing force field parameters were generated using the *antechamber* and *parmchk* utilities of the AMBER suite (16.0). All structure visualization was performed in PyMOL (https://pymol.org/).

**Automated fragmentation quantum mechanics/molecular mechanics chemical-shift calculations.** We generated mono-nucleoside models of m⁶A with the N1-C6-N6-methyl carbon dihedral angle ranging from 0° to 360° in steps of 20° (*syn* conformation is 0°, whereas *anti* conformation is 180°). Coordinates of the m⁶A residue were derived from Aduri et al.[72]. We subjected the various mono-nucleoside models and all the RNA duplex MD ensembles (each with $N = 100$) to QM/MM chemical-shift calculations using a fragmentation procedure as described previously[73]. The parameters of geometric minimization for RNA structures were described in a prior study[74]. For all the RNA duplex ensembles, the chemical-shift calculations were solely focused on A6 and U13 residues in dsGGACU; therefore, each conformer in the RNA duplex ensembles was broken into only two quantum fragments centered on A6 or U13, respectively, whereas for all the mono-nucleoside models, each quantum fragment was the single mono-nucleoside. We then used a distribution of point charges on the fragment surface to represent the effects of RNA that is outside the quantum fragment and solvent[75]. The local dielectric ε value was set to be 1, 4, and 80 for RNA inside the quantum fragment, RNA outside the quantum fragment, and RNA outside the solvent, respectively. We then performed the GIAO chemical-shift calculations for each quantum fragment with the OLYP functional and the pcSseg-0 basis set, using demon-2k program (http://www.demon-software.com/public_html/download.html). Reference shieldings were computed for TMS and nitromethane at the same level of theory.

**Free energy decomposition along the CS pathway.** The free energy of annealing the methylated duplex can be decomposed into two steps (CS pathway):

$$\text{Step 1}: \ \text{ssRNA}^{syn} \rightleftharpoons \text{ssRNA}^{anti} \quad (12)$$

$$\text{Step 2}: \ \text{ssRNA}^{anti} + \text{ss2} \rightleftharpoons \text{dsRNA}^{anti} \quad (13)$$

$k_1$ and $k_{-1}$ were determined from two-state fits or temperature dependence of the RD data (see 'Determining the methylamino isomerization rate constants from temperature-dependent RD measurements for methylated ssRNA and dsRNA' section above):

$$\Delta G^\circ_{iso,ss} = -RT\ln\left(\frac{k_1}{k_{-1}}\right) \quad (14)$$

The apparent free energy of annealing methylated dsRNA was determined using:

$$\Delta G^{\circ app}_{anneal,m^6A} = -RT\ln\left(\frac{[\text{ssRNA}^{syn}][\text{ss2}]}{[\text{dsRNA}^{anti}]}\right) \quad (15)$$

in which the concentrations of the relevant species were measured based on two-state fits of the RD data[21]:

$$[ssRNA^{anti}] = \frac{[ssRNA^{total}] \times k_1}{k_1 + k_{-1}} \quad (16)$$

$$[ssRNA^{syn}] = \frac{[ssRNA^{total}] \times k_{-1}}{k_1 + k_{-1}} \quad (17)$$

$$[ssRNA^{total}] = C_t \times p_{ss} \quad (18)$$

$$[dsRNA^{anti}] = C_t \times p_{GS} \quad (19)$$

$$[ss2] = [ss2]_{total} - [dsRNA^{anti}] - [dsRNA^{syn}] \quad (20)$$

$$[dsRNA^{syn}] = C_t \times p_{ES} \quad (21)$$

$p_{ss}$ and $p_{GS}$ are the populations of the ssRNA^{total} (ssRNA^{syn} + ssRNA^{anti}) and dsRNA^{anti} species obtained from the RD data. [ss2]_{total} is the total complementary strand concentration. Note that at $T = 65$ °C, dsRNA^{syn} has a negligible contribution to RD profiles ([dsRNA^{anti}] = 0), while at $T = 55$ °C, dsRNA^{syn} population ($p_{ES}$) was obtained from a three-state fit of the m⁶A-C2 CEST data for dsGGACU^{m⁶A}. Also

note that the $\Delta G^{\circ}{}_{anneal,m^6A}^{app}$ here differs slightly (by ~0.1 kcal/mol) from the prior study[21], where ssRNA$^{syn}$ and ssRNA$^{anti}$ were not distinguished.

The free energy of annealing ssRNA$^{anti}$ is given by:

$$\Delta G^{\circ}_{anneal,anti} = \Delta G^{\circ}{}_{anneal,m^6A}^{app} - \Delta G^{\circ}_{iso,ss} \tag{22}$$

$$k_{off,anti} = \frac{k_{on,anti}}{e^{\frac{\Delta G^{\circ}_{anneal,anti}}{-RT}}} \tag{23}$$

$$\Delta\Delta G^{\circ}_{anneal,anti} = \Delta G^{\circ}_{anneal,anti} - \Delta G^{\circ}_{anneal,A} \tag{24}$$

At $T = 55\,°C$, $\Delta\Delta G^{\circ}_{anneal,anti} = 0.5 \pm 0.2$ kcal/mol, and the m$^6$A methyl group in *anti* conformation slightly destabilizes the duplex, whereas it stabilized it by a comparable amount at $T = 65\,°C$ ($\Delta\Delta G^{\circ}_{anneal,anti} = -0.5 \pm 0.2$ kcal/mol).

**B–M simulations and constrained fits.** When dealing with three- or four-state exchange, there is always a danger of overfitting the RD data. For this reason, we initially performed simulations in which all of the relevant kinetic rate constants, populations, $\Delta\omega$, $R_1$, and $R_2$ of the different species were approximated to values measured experimentally using the appropriate RNA constructs (Supplementary Figs. 2b and 6a and Supplementary Table 1). These values were then used in a three- or four-state simulation to simulate CEST profiles without any adjustable parameters. We then performed constrained fits in which the parameters (population, rate constants, $\Delta\omega$, $R_1$, and $R_2$) were allowed to float by an amount determined by the experimentally measured uncertainty (1 s.d.).

The simulations and constrained fits were performed by numerically integrating the appropriate B–M equations[34]. Briefly, the simulations were performed by directly predicting RD profiles for a given set of exchange parameters that are defined below. In the constrained fitting, the same numerical integration was used to fit exchange parameters applying specific constraints as detailed below.

*Three-state CS simulations and constrained fits for the* dsA6DNA$^{m^6A}$ *RD data measured at* $T = 65\,°C$. These analyses used the following input exchange parameters:

1. $k_1$ and $k_{-1}$ were obtained from the temperature-dependent RD measurements on ssGGACU$^{m^6A}$ (Supplementary Fig. 2c).
2. $k_{off,anti}$ was assumed equal to $k_{off}$ measured for the unmethylated dsGGACU and $k_{on,anti}$ was obtained from the energetic decomposition described above (Eq. 23).
3. The longitudinal ($R_1$) and transverse ($R_2$) relaxation rate constants for all three species (ssRNA$^{syn}$, ssRNA$^{anti}$, and dsRNA$^{anti}$) were obtained from two-state fits of the CEST RD data probing duplex melting at $T = 65\,°C$[21]. $R_1$(ssRNA$^{anti}$) = $R_1$(ssRNA$^{syn}$) = $R_1$(dsRNA$^{anti}$) = $R_{1,GS}$ = $R_{1,ES}$. $R_2$(ssRNA$^{anti}$) = $R_2$(ssRNA$^{syn}$) = $R_{2,ES}$. $R_2$(dsRNA$^{anti}$) = $R_{2,GS}$.
4. The equilibrium populations $p_{(ssRNA^{syn})}$, $p_{(ssRNA^{anti})}$, and $p_{(dsRNA^{anti})}$ were obtained from kinetic simulations (see differential equations below) that were sufficiently long to ensure equilibrium. The same equilibrium populations were obtained from analytical expressions outlined in ref. [76]:

$$\frac{d[ssRNA^{syn}]}{dt} = -k_1[ssRNA^{syn}] + k_{-1}[ssRNA^{anti}] \tag{25}$$

$$\frac{d[ssRNA^{anti}]}{dt} = k_1[ssRNA^{syn}] - k_{-1}[ssRNA^{anti}] - k_{on,anti}[ssRNA^{anti}][ss2] + k_{off,anti}[dsRNA^{anti}] \tag{26}$$

$$\frac{d[dsRNA^{anti}]}{dt} = k_{on,anti}[ssRNA^{anti}][ss2] - k_{off,anti}[dsRNA^{anti}] \tag{27}$$

$$\frac{d[ss2]}{dt} = -k_{on,anti}[ssRNA^{anti}][ss2] + k_{off,anti}[dsRNA^{anti}] \tag{28}$$

5. $\Delta\omega$ of ssRNA$^{syn}$ and ssRNA$^{anti}$ for C2: $\Delta\omega_{ss,syn} = \omega_{ss,syn} - \omega_{ds,anti}$, in which $\omega_{ss,syn} = \omega_{ss} - \frac{p_{(ssRNA^{syn})}}{p_{(ssRNA^{syn})}+p_{(ssRNA^{anti})}} \times \Delta\omega_{ss,anti-syn}$. $\Delta\omega_{ss,anti} = \omega_{ss,anti} - \omega_{ds,anti}$, in which $\omega_{ss,anti} = \omega_{ss} + \frac{p_{(ssRNA^{anti})}}{p_{(ssRNA^{syn})}+p_{(ssRNA^{anti})}} \times \Delta\omega_{ss,anti-syn}$. $\omega_{ss}$ and $\omega_{ds,anti}$ were obtained from 2D HSQC spectra (Supplementary Fig. 1) and $\Delta\omega_{ss,anti-syn}$ was obtained from ssGGACU$^{m^6A}$ RD measurements at $T = 25\,°C$ and was assumed to be temperature independent, as supported by the data collected in this study (Supplementary Fig. 2, Supplementary Table 1). Since C8 is not sensitive to methylamino isomerization (Supplementary Fig. 2), $\Delta\omega_{ss,anti} = 0$, while $\Delta\omega_{ss,syn}$ is obtained from the two-state fit of the CEST RD data probing duplex melting at $T = 65\,°C$[21].

The above parameters were fixed to simulate the CEST profiles using a three-state B–M equation[34]. For the constrained three-state fit, the ratio (but not absolute magnitude) of $k_{on,anti}$ to $k_{off,anti}$ was constrained to preserve the free energy of the hybridization step. All other parameters (population, $k_1$, $k_{-1}$, $\Delta\omega$, $R_1$, and $R_2$ for all

species) were allowed to float by an amount determined by the uncertainty (1 s.d.). When possible, global constrained three-state B–M fits were carried out on both m$^6$A C8 and C2 CEST data (Fig. 2f). $\chi^2_{red}$ was calculated to assess the goodness of fitting[18].

*Three-state IF simulations and constrained fits for the* dsGGACU$^{m^6A}$ *RD data were measured at* $T = 65$ *and* $55\,°C$. These analyses used the following input exchange parameters:

1. $k_2$, $k_{-2}$, $k_{on,syn}$, $k_{off,syn}$, C2 $\Delta\omega_{ds,syn}$, C2 $R_1$(dsRNA$^{syn}$) = $R_1$(dsRNA$^{anti}$) = $R_{1,GS}$, and $R_2$(dsRNA$^{syn}$) = $R_2$(dsRNA$^{anti}$) = $R_{2,GS}$ were obtained from a three-state fit to the dsGGACU$^{m^6A}$ m$^6$A-C2 RD data (Fig. 3a and Supplementary Table 3) using the triangular topology at $T = 55\,°C$ or from RD measurements done on the hairpin constructs at $T = 65\,°C$ (Supplementary Fig. 6a). C8 $\Delta\omega_{ds,syn} = 0$ because C8 is not sensitive to methylamino isomerization (Supplementary Fig. 6a). C8 $R_1$(dsRNA$^{syn}$) = $R_1$(dsRNA$^{anti}$) = $R_{1,GS}$, and $R_2$(dsRNA$^{syn}$) = $R_2$(dsRNA$^{anti}$) = $R_{2,GS}$ were obtained from a two-state fit to the dsGGACU$^{m^6A}$ m$^6$A-C8 RD data (Supplementary Table 1), $\Delta\omega_{ss,syn}$ is obtained from the two-state fit of the CEST RD data probing duplex melting at $T = 65$ and $55\,°C$[21].
2. The equilibrium populations $p_{(ssRNA^{syn})}$, $p_{(dsRNA^{syn})}$, $p_{(dsRNA^{anti})}$ were obtained from kinetic simulations (see differential equations below) that were sufficiently long to ensure equilibration:

$$\frac{d[ssRNA^{syn}]}{dt} = k_{off,syn}[dsRNA^{syn}] - k_{on,syn}[ssRNA^{syn}][ss2] \tag{29}$$

$$\frac{d[dsRNA^{syn}]}{dt} = -k_{off,syn}[dsRNA^{syn}] + k_{on,syn}[ssRNA^{syn}][ss2] + k_2[dsRNA^{anti}] - k_{-2}[dsRNA^{syn}] \tag{30}$$

$$\frac{d[dsRNA^{anti}]}{dt} = k_{-2}[dsRNA^{syn}] - k_2[dsRNA^{anti}] \tag{31}$$

$$\frac{d[ss2]}{dt} = k_{off,syn}[dsRNA^{syn}] - k_{on,syn}[ssRNA^{syn}][ss2] + k_{off,anti}[dsRNA^{anti}] \tag{32}$$

The same approach was used to simulate/fit CEST profiles for the IF pathway as described in the previous section.

*Four-state CS + IF simulations and constrained fits for the* dsGGACU$^{m^6A}$ *RD data at* $T = 55\,°C$. These analyses used the following input exchange parameters:

1. All of the exchange parameters related to the CS pathway ($k_1$, $k_{-1}$, $k_{on,anti}$, $k_{off,anti}$, $\Delta\omega_{ss,syn}$, $\Delta\omega_{ss,anti}$, $R_1$(dsRNA$^{anti}$), $R_1$(ssRNA$^{syn}$), $R_1$(dsRNA$^{anti}$), $R_2$(dsRNA$^{anti}$), $R_2$(ssRNA$^{syn}$), and $R_2$(dsRNA$^{anti}$)) and the IF pathway ($k_{-2}$, $k_2$, $k_{on,syn}$, $k_{off,syn}$, $R_1$(dsRNA$^{syn}$), and $\Delta\omega_{ds,syn}$) were obtained as described in the previous sections for the three-state CS and IF analysis, respectively.
2. The population of all four species was obtained from four-state kinetic simulations using the eight rate constants ($k_1$, $k_{-1}$, $k_{on,anti}$, $k_{off,anti}$, $k_{-2}$, $k_2$, $k_{on,syn}$, $k_{off,syn}$) based on the CS + IF model (see differential equations below). The same equilibrium populations were obtained from analytical expressions outlined in ref. [76]:

$$\frac{d[ssRNA^{syn}]}{dt} = -k_1[ssRNA^{syn}] + k_{-1}[ssRNA^{anti}] + k_{off,syn}[dsRNA^{syn}] - k_{on,syn}[ssRNA^{syn}][ss2] \tag{33}$$

$$\frac{d[ssRNA^{anti}]}{dt} = k_1[ssRNA^{syn}] - k_{-1}[ssRNA^{anti}] - k_{on,anti}[ssRNA^{anti}][ss2] + k_{off,anti}[dsRNA^{anti}] \tag{34}$$

$$\frac{d[dsRNA^{syn}]}{dt} = -k_{off,syn}[dsRNA^{syn}] + k_{on,syn}[ssRNA^{syn}][ss2] + k_2[dsRNA^{anti}] - k_{-2}[dsRNA^{syn}] \tag{35}$$

$$\frac{d[dsRNA^{anti}]}{dt} = k_{on,anti}[ssRNA^{anti}][ss2] - k_{off,anti}[dsRNA^{anti}] + k_{-2}[dsRNA^{syn}] - k_2[dsRNA^{anti}] \tag{36}$$

$$\frac{d[ss2]}{dt} = k_{off,syn}[dsRNA^{syn}] - k_{on,syn}[ssRNA^{syn}][ss2] - k_{on,anti}[ssRNA^{anti}][ss2] + k_{off,anti}[dsRNA^{anti}] \tag{37}$$

The exchange parameters described above were then used to simulate the CEST profile using the four-state B–M equation (see below)[62]:

$$
\frac{d}{dt}
\begin{pmatrix}
I_{GSx} \\
I_{GSy} \\
I_{GSz} \\
I_{ES1x} \\
I_{ES1y} \\
I_{ES1z} \\
I_{ES2x} \\
I_{ES2y} \\
I_{ES2z} \\
I_{ES3x} \\
I_{ES3y} \\
I_{ES3z}
\end{pmatrix}
=
$$

$$
\begin{pmatrix}
-R_{2,GS}-k_{12}-k_{54} & -\Omega_{GS} & \omega & k_{21} & 0 & 0 & 0 & 0 & 0 & k_{45} & 0 & 0 \\
\Omega_{GS} & -R_{2,GS}-k_{12}-k_{54} & 0 & 0 & k_{21} & 0 & 0 & 0 & 0 & 0 & k_{45} & 0 \\
-\omega & 0 & -R_{1,GS}-k_{12}-k_{54} & 0 & 0 & k_{21} & 0 & 0 & 0 & 0 & 0 & k_{45} \\
k_{12} & 0 & 0 & -R_{2,ES1}-k_{23}-k_{21} & -\Omega_{ES1} & \omega & k_{32} & 0 & 0 & 0 & 0 & 0 \\
0 & k_{12} & 0 & \Omega_{ES1} & -R_{2,ES1}-k_{23}-k_{21} & 0 & 0 & k_{32} & 0 & 0 & 0 & 0 \\
0 & 0 & k_{12} & -\omega & 0 & -R_{1,ES1}-k_{23}-k_{21} & 0 & 0 & k_{32} & 0 & 0 & 0 \\
0 & 0 & 0 & k_{23} & 0 & 0 & -R_{2,ES2}-k_{32}-k_{34} & -\Omega_{ES2} & \omega & k_{43} & 0 & 0 \\
0 & 0 & 0 & 0 & k_{23} & 0 & \Omega_{ES2} & -R_{2,ES2}-k_{32}-k_{34} & 0 & 0 & k_{43} & 0 \\
0 & 0 & 0 & 0 & 0 & k_{23} & -\omega & 0 & -R_{1,ES2}-k_{32}-k_{34} & 0 & 0 & k_{43} \\
k_{54} & 0 & 0 & 0 & 0 & 0 & k_{34} & 0 & 0 & -R_{2,ES3}-k_{43}-k_{45} & -\Omega_{ES3} & \omega \\
0 & k_{54} & 0 & 0 & 0 & 0 & 0 & k_{34} & 0 & \Omega_{ES3} & -R_{2,ES3}-k_{43}-k_{45} & 0 \\
0 & 0 & k_{54} & 0 & 0 & 0 & 0 & 0 & k_{34} & -\omega & 0 & -R_{1,ES3}-k_{43}-k_{45}
\end{pmatrix}
\begin{pmatrix}
I_{GSx} \\
I_{GSy} \\
I_{GSz} \\
I_{ES1x} \\
I_{ES1y} \\
I_{ES1z} \\
I_{ES2x} \\
I_{ES2y} \\
I_{ES2z} \\
I_{ES3x} \\
I_{ES3y} \\
I_{ES3z}
\end{pmatrix}
$$

$$
+
\begin{pmatrix}
0 \\
0 \\
R_{1,GS}I_{GSz,eq} \\
0 \\
0 \\
R_{1,ES1}I_{ES1z,eq} \\
0 \\
0 \\
R_{1,ES2}I_{ES2z,eq} \\
0 \\
0 \\
R_{1,ES3}I_{ES3z,eq}
\end{pmatrix}
\tag{38}
$$

{GS/ES$_i$}{x/y/z} ($i = 1, 2, 3$) denotes the magnetization of the GS or ESs in the specified direction. $R_{2,GS}$, $R_{2,ES1}$, $R_{2,ES2}$, and $R_{2,ES3}$ are the transverse relaxation rate constants for the GS (dsRNA$^{anti}$), ES1 (dsRNA$^{syn}$), ES2 (ssRNA$^{syn}$), and ES3 (ssRNA$^{anti}$), respectively. $R_{1,GS}$, $R_{1,ES1}$, $R_{1,ES2}$, and $R_{1,ES3}$ are corresponding longitudinal relaxation rate constants. $\omega$ is the RF field power; $k_{\{ij\}}$ and $k_{\{ji\}}$ are the forward and backward rate constants of reactions shown in Fig. 5a. Specifically, $k_{12} = k_2$, and $k_{21} = k_{-2}$ are the forward and backward rate constants of methylamino isomerization in dsRNA. $k_{23} = k_{off,syn}$, $k_{32} = k_{on,syn}$[ss2]. $k_{34} = k_1$ and $k_{43} = k_{-1}$ are the forward and backward rate constants of methylamino isomerization in ssRNA. $k_{45} = k_{on,anti}$[ss2] and $k_{54} = k_{off,anti}$. $I_{\{GS/ES_i\}z,eq}$ ($i = 1, 2, 3$) denotes the longitudinal magnetization of the GS or ESs at the start of the experiment. $\omega_i$ ($i = 1, 2, 3, 4$) are the offset frequencies of the GS, or ES resonances in the rotating frame of the RF field[61].

We carried out two independent constrained four-state fits at $T = 55\,°C$ that differ with regards to how $k_{on,syn}$ and $k_{off,syn}$ were defined. In one case, $k_{on,syn}$ was assumed to be equal to the $k_{ss \to ES}$ rate constant obtained from a three-state fit to the CEST data measured for dsGGACU$^{m6A}$ at $T = 55\,°C$ (Fig. 3b) using the triangular topology. Note that this is an approximation since the ssRNA represents the major ssRNA$^{syn}$ and minor ssRNA$^{anti}$ species in fast exchange. $k_{off,syn}$ was then calculated by closing the thermodynamic cycle:

$$
\Delta G^{\circ}_{anneal,syn} = \Delta G^{\circ\,app}_{anneal,m6A} - \Delta G^{\circ}_{iso,ds}
\tag{39}
$$

$$
\Delta G^{\circ}_{iso,ds} = -RT \ln\left(\frac{k_{-2}}{k_2}\right)
\tag{40}
$$

$$
k_{off,syn} = \frac{k_{on,syn}}{e^{\frac{\Delta G^{\circ}_{anneal,syn}}{-RT}}}
\tag{41}
$$

All other exchange parameters were then allowed to float by an amount determined by the experimental uncertainty (one standard deviation). In the second case, only the ratio (but not absolute magnitude) of $k_{on,syn}$ to $k_{off,syn}$ was constrained to preserve the free energy of the hybridization step. The fitted $k_{on,syn}$ and $k_{off,syn}$ values were similar using these two independent methods. The results from the second method were reported in Fig. 5a and Supplementary Table 2. When possible, global constrained four-state B–M fits were carried out on both m6A C8 and C2 CEST data. $\chi^2_{red}$ was calculated to assess the goodness of fitting[18].

*Four-state constrained fits for the CS + IF model for* dsGGACU$^{m6A}$ *at* T = 65 °C. Because the dsRNA$^{syn}$ ES was not directly detected at $T = 65\,°C$, the RD data were analyzed as described for $T = 55\,°C$, with the exception that $k_2$ and $k_{-2}$ were measured in hpGGACU$^{m6A}$ at $T = 65\,°C$ using $R_{1\rho}$ RD (Supplementary Fig. 6a), $k_{on,syn}$ was assumed to be equal to $k_{on}/20$. This 20-fold slowdown in annealing of ssRNA$^{syn}$ relative to unmethylated ssRNA was observed for dsGGACU$^{m6A}$ at $T = 55\,°C$. $k_{off,syn}$ was then calculated by closing the thermodynamic cycle (Eq. 37). Similar results were obtained when assuming $k_{off,syn}$ is equal to $k_{off} \times 80$ as observed for dsGGACU$^{m6A}$ at $T = 55\,°C$, and closing the cycle (Eq. 37) to calculate $k_{on,syn}$.

*Four-state constrained fits for the CS + IF model for* dsHCVm6A *and* dsA6DNA$^{m6A}$. RD data measured for dsHCVm6A and dsA6DNA$^{m6A}$ were analyzed in a similar manner as described in the previous sections.

1. $k_1$, $k_{-1}$ and $k_2$, $k_{-2}$ were assumed to be the same as those measured in GGACU$^{m6A}$ constructs using temperature-dependent RD measurements (Supplementary Figs. 2c and 6d).

2. $R_1$(ssRNA$^{anti}$) = $R_1$(ssRNA$^{syn}$) = $R_1$(dsRNA$^{anti}$) = $R_{1,GS}$ = $R_{1,ES}$. $R_2$(dsRNA$^{anti}$) = $R_2$(ssRNA$^{syn}$) = $R_{2,ES}$. $R_2$(dsRNA$^{anti}$) = $R_{2,GS}$. $R_{1,ES}$ and $R_{2,GS}$ were obtained from a two-state fit to the RD data probing duplex melting (Supplementary Table 1).

3. $\Delta\omega_{ss,anti} = \Delta\omega_{ds,syn} = 0$ for A11-C8 in dsHCVm6A since A11 is not the m6A site. $\Delta\omega_{ss,syn}$ was assumed to be equal to the $\Delta\omega$ value for A11-C8 in ssRNA obtained from a two-state fit of the A11-C8 RD data[21].

4. $\Delta\omega_{ss,syn}$ and $\Delta\omega_{ss,anti}$ for m6A16-C2 in dsA6DNA$^{m6A}$ were determined as described in CS three-state simulation for dsGGACU$^{m6A}$ at $T = 65\,°C$, assuming $\Delta\omega_{ss,anti-syn}$ of ssA6DNAm6A is the same as that of ssGGACU$^{m6A}$. $\Delta\omega_{ds,syn}$ was assumed to be equal to that measured for hpGGACU$^{m6A}$ at $T = 55\,°C$ (Supplementary Table 1).

**Flux calculations**. Flux through the of CS ($F_{CS}$) and IF ($F_{IF}$) pathways was calculated as the harmonic mean of the forward rates along the CS and IF pathways[27]:

$$
F_{CS} = \left(\frac{1}{k_1[ssRNA^{syn}]} + \frac{1}{k_{on,anti}[ssRNA^{anti}][ss2]}\right)^{-1}
\tag{42}
$$

$$
F_{IF} = \left(\frac{1}{k_{on,syn}[ssRNA^{syn}][ss2]} + \frac{1}{k_{-2}[dsRNA^{syn}]}\right)^{-1}
\tag{43}
$$

All concentrations are equilibrium concentrations obtained using constrained four-state fit of CEST data (Fig. 5c) or CS + IF kinetic modeling.

**Model to predict apparent $k_{on}$ and $k_{off}$ for methylated RNA/DNA duplexes and TAR**. The four-state CS + IF model was used to simulate time traces describing the evolution of all four species as a function of time starting from 100% ssRNA$^{syn}$ at $t = 0$. Similar results were obtained when performing simulations starting with an equilibrium population of ssRNA$^{syn}$ ($k_{-1}/(k_1 + k_{-1})$) and ssRNA$^{anti}$ ($k_1/(k_1 + k_{-1})$). $k_1$, $k_{-1}$, $k_{-2}$, and $k_2$ were all assumed equal to the corresponding values measured for ssGGACU$^{m6A}$ and dsGGACU$^{m6A}$ at the appropriate temperature based on the temperature-dependent RD measurements (Supplementary Figs. 2c and 6d). $k_{off,anti}$ was assumed to be equal to $k_{off}$, and $k_{on,anti}$ was deduced from closing the thermodynamic cycle (Eq. 23). $k_{on,syn}$ and $k_{off,syn}$ were obtained using two different approaches and yielded similar predictions for the apparent $k_{on}$ and $k_{off}$ for methylated RNA/DNA duplexes and TAR. In one case, $k_{on,syn} = k_{on}/20$ and $k_{off,syn}$ was deduced from closing the thermodynamic cycle (Eq. 37). Alternatively, $k_{off,syn} = k_{off} \times 80$ and $k_{on,syn}$ was deduced from closing the thermodynamic cycle (Eq. 37). The predictions shown in Fig. 6a

were obtained using the former approach. $k_{on,m^6A}^{app}$ and $k_{off,m^6A}^{app}$ were obtained by fitting simulated time course of $[dsRNA^{syn}] + [dsRNA^{anti}]$ at multiple time points to numerical solutions of Eqs. (40) and (41) for a two-state hybridization model ss1 + ss2 $\Delta ds$, $k_{on,m^6A}^{app}$, and $k_{off,m^6A}^{app}$ are the annealing and melting constants, respectively:

$$\frac{d[ds]}{dt} = k_{on,m^6A}^{app}[ss1][ss2] - k_{off,m^6A}^{app}[ds] \tag{44}$$

$$\frac{d[ss1]}{dt} = \frac{d[ss2]}{dt} = -k_{on,m^6A}^{app}[ss1][ss2] + k_{off,m^6A}^{app}[ds] \tag{45}$$

Similar results were obtained when fitting simulated time course of $[dsRNA^{anti}]$ only. However, it should be noted that for certain kinetic regimes outside those examined here, particularly when $k_{on,syn}$ is ultra-fast, there can be a substantial accumulation of the $dsRNA^{syn}$. In this scenario, the system is poorly defined with the apparent two-state approximation and separate rate constants are needed to describe the evolution of all species. In addition, similar results were obtained from fitting the traces to the appropriate two-state second-order kinetic equation (see ref. [77]). Finally, similar results were obtained when simulating m6A-C8 RD profiles using the four-state CS + IF model together with exchange parameters ($\Delta\omega$, $R_1$, and $R_2$ values for all species) derived from the dsGGACU$^{m^6A}$ 55 °C m6A-C8 CEST data, and then fitting the data to a two-state model. Note that C8 was used the probe instead of C2 because the two-state fit results vary depending on the three $\Delta\omega$ values used in the C2 CEST simulation. On the other hand, varying the one $\Delta\omega$ value used in the C8 CEST simulation does not affect the two-state fit results. As the choice of exchange parameters ($R_1$ and $R_2$ values) had a minor effect on the two-state fit results, we show results from the kinetic simulations in Fig. 6a and that from the two-state fitting to the simulated C8 RD data in Supplementary Fig. 9b.

A similar approach was used to compute the apparent $k_{forward}$ and $k_{backward}$ rate constants for methylated TAR, except that $k_1$, $k_{-1}$ were assumed to be equal to the values measured for m6AMP, which is a better mimic of the environment of the flipped out and unstacked A35 in TAR than ssRNA. Apparent $k_{forward}$ and $k_{backward}$ rate constants were obtained by fitting simulated time course of [ES] at multiple time points to the equation $[ES] = A(1 - e^{-k_{ex}t})$, where $A$ is a pre-exponential factor. Note that for the energetics decomposition and kinetic simulations of TAR, the [ss2] term in all equations above was removed since the TAR conformational transition is a first-order reaction.

### Predict m6A-induced slowdown of DNA hybridization in the mouse genome.
We used our four-state CS + IF model to predict the hybridization kinetics for 12-mer DNA duplex representing 5951 m6A sites in the mouse genome[9], in which m6A was positioned at the sixth nucleotide. $k_{on}$ of unmethylated DNA was predicted as described previously[45] (http://nablab.rice.edu/nabtools/kinetics.html). The free energy ($\Delta G_{anneal,A}^\circ$) of each sequence was predicted using the MELTING program (https://www.ebi.ac.uk/biomodels-static/tools/melting/). $k_{off} = \frac{k_{on}}{e^{\frac{\Delta G_{anneal,A}^\circ}{-RT}}}$. In all cases, the thermodynamic destabilization of the duplex by m6A ($\Delta\Delta G_{anneal,m^6A}^\circ$) was assumed to be 1 kcal/mol based on prior studies[12,78] and our measurements (Supplementary Table 4). $\Delta G_{anneal,m^6A}^{\circ\,app}$ was obtained from $\Delta G_{anneal,m^6A}^{\circ\,app} = \Delta G_{anneal,A}^\circ + \Delta\Delta G_{anneal,m^6A}^{\circ\,app}$. $k_{on}$, $k_{off}$, and $\Delta G_{anneal,m^6A}^{\circ\,app}$ were then used as inputs to predict $k_{on,m^6A}^{app}$ and $k_{off,m^6A}^{app}$ as described in the previous sections. The concentration of dsDNA was assumed to be 1 mM and $T = 37$ °C. We also used this approach to predict the impact of m6A on RNA hybridization kinetics at $T = 37$ °C using rate constants for hybridization of unmethylated RNA reported previously[22] at $T = 37$ °C and assuming that m6A destabilizes dsRNA by 1 kcal/mol[12]. m6A was predicted to slow $k_{on}$ by ~5-fold while having a minor effect (<2-fold) on $k_{off}$, consistent with our measurements at higher temperatures.

### Survey of single H-bonded A–U bps in PDB structures.
To identify singly H-bonded A–U bp conformations that mimic the m6(syn)A···U ES, we conducted a structural survey of the RCSB PDB[79]. All X-ray (with resolution ≤ 3.0 Å) and NMR biological assemblies containing RNA molecules (including naked RNA, RNA protein complex, etc.) were downloaded from RCSB PDB on Aug 2017 and processed by X3DNA-DSSR[80] to generate a searchable database containing RNA structural information. Potential candidates of single H-bonded A–U bp were identified by applying the following filters in the database: (1) A–U bps are unmethylated; (2) the Leontis–Westhof (LW) classification[81] is "cWW"; (3) both A and U are not in syn conformation at glycosidic bond; (4) A–U bps contain A(N1)–U(N3) H-bond (distance between A(N1) and U(N3) is <3.5 Å) but do not contain A(N6)–U(O4) H-bond (distance between A(N6) and U(O4) is >3.5 Å). We then manually inspected all the single H-bonded A–U bps, removed misregistered bps, and classified the structure context of all the resulting bps into the following categories (Supplementary Fig. 7e):

1. Junction: A–U bp that is next to an internal bulge, a mismatch or an apical loop.
2. Junction-1/2/3: 1/2/3 bp away from the junction.
3. Tertiary: involved in tertiary interactions.
4. Terminal: at terminal ends.
5. Terminal-1/2/3: 1/2/3 bp away from the terminal end.
6. Duplex: A–U bp at the canonical duplex context

**Reporting summary**. Further information on research design is available in the Nature Research Reporting Summary linked to this article.

## Data availability
The data supporting the findings of this study are available from the corresponding authors upon reasonable request. The NMR $R_{1\rho}$ (Fig. 2c and Supplementary Figs. 2a, b, 5, 6a, and 10b, c, d), CEST (Figs. 2c, f, 3a, c, 4b–d, and 5b and Supplementary Figs. 2b, 3a–c, 4a, c–e, 5, 6a, and 9a, d) and imino exchange (Supplementary Fig. 8c, d) data as well as kinetic simulation and prediction results (Fig. 6a and Supplementary Fig. 9b) generated in this study are provided at https://github.com/alhashimilab/m6A_hybridization_kinetics (https://doi.org/10.5281/zenodo.5106694)[82]. The force field parameters for m6A and m6$_2$A used in MD simulations and PDB files of these structures that were submitted to the DFT calculations (Fig. 4g and Supplementary Fig. 7c) are provided at https://github.com/alhashimilab/m6A_ES (https://doi.org/10.5281/zenodo.5099581)[83]. The results of PDB (RCSB Protein Data Bank) survey for singly H-bond A–U bps (Supplementary Fig. 7f) are provided at https://github.com/alhashimilab/Singly_HB_AU (https://doi.org/10.5281/zenodo.5099558)[84]. The DNA m6A sites (Fig. 6f) used in this study were reported in a prior study[9]. See Supplementary Table 5 of the cited paper (https://www.nature.com/articles/nature17640#Sec26). Source data for Figs. 2d, 4g, and 6a, c, e and Supplementary Figs. 2c, 4b, 6d, 7c, d, f, 8c, d, 9b, c, 10b, and 11a–d are provided with this paper.

## Code availability
In-house Python scripts used to perform CEST fitting, kinetic simulations, and predictions (duplex hybridization, TAR conformational transitions, and genome-wide DNA hybridization prediction) are provided at https://github.com/alhashimilab/m6A_hybridization_kinetics (https://doi.org/10.5281/zenodo.5099562)[82].

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

## Acknowledgements

We thank members of the Al-Hashimi laboratory for their assistance and critical comments on the manuscript. We would like to thank Yanjiao Wang and Prof. Xue Yi (Tsinghua University) for the help with the $^1$H CEST experiment, Prof. Terrence Oas (Duke University) for advice about kinetic simulations and calculations, and Prof. Qi Zhang for providing the 2D [$^{13}$C, $^1$H] CEST pulse sequence based on which the methyl CEST sequence was derived. This work was supported by the US National Institute for General Medical Sciences (1R01GM132899) and US National Institute of Health (R01GM089846) to H.M.A., the Austrian Science Fund (FWF, project P30370 and P32773) and the Austrian Research Promotion Agency FFG (West Austrian BioNMR, 858017) to C.K. and the National Institute for Allergy and Infectious Diseases (U54 AI150470) to D.A.C.

## Author contributions

B.L. and H.M.A. conceived the project and experimental design. B.L. prepared NMR samples, performed NMR experiments, and analyzed NMR data with the help from H.S., A.R. and C.-C.C., F.N., K.A.E. and C.K. prepared ($^{13}$CH$_3$)-m$^6$A RNA phosphoramidite and $^{13}$C8,$^{13}$C2-labeled m$^6$dA phosphoramidite. B.L. performed kinetic simulations and predictions. H.S. performed proton CEST and imino proton exchange experiments. A.R. performed MD simulations. H.S. and D.A.C. performed Automated fragmentation quantum mechanics/molecular mechanics (AF-QM/MM) chemical-shift calculations. B.L. and H.S. performed the PDB survey. H.M.A. and B.L. wrote the manuscript with critical input from H.S. and A.R.

## Competing interests

H.M.A. is an advisor to and holds an ownership interest in Nymirum, an RNA-based drug discovery company. C.K. is an advisor to and holds an ownership interest in INNotope, a company providing RNA stable isotope labeling products. The remaining authors declare no competing interests.
