## [Peer Review File · Nature Communications]

REVIEWER COMMENTS

Reviewer #1 (Remarks to the Author):

This paper describes a detailed relaxation dispersion NMR investigation of the role of m6A methylation in regulating strand formation and melting. Using sophisticated NMR dynamics methods, a series of nucleic acid constructs, different temperatures and labeling that stabilizes conformational preferences, a model is described consisting of separate induced fit and conformational selection pathways that explains the kinetics of duplex formation/melting. Notably, the intermediate formed in the induced fit pathway is one where the syn conformation is retained in the duplex state – the first time that such a conformation has been observed. However, the authors provide convincing evidence that indeed this is in fact the case. This has potential implications for recognition of nucleic acids by cognate proteins and further enhances our knowledge of the different structures that these molecules can form.

The work represents an important contribution to the literature and a stunning example of how to dissect what appears to be a complex 4 state pathway. I have only 1 simple question. Could the authors provide an intuitive explanation for why at 65oC the syn double strand conformation is much lower than at 55oC, that is, what influences the DH of the process so significantly?

Reviewer #2 (Remarks to the Author):

Liu et al. present a study of a quantitative model to predict kinetic behaviors of nucleic acid consisting of m6A methylation in hybridization and conformational transitions. The m6A methylation is an important posttranscriptional modification in the regulation of gene expression. This study shows that m6A modulates the annealing/melting behavior of nucleic acids through syn/anti-conversion and conformational selection pathways. The study is highly significant; the approach and data analysis appear to be rigorous. After addressing the following concerns, the manuscript is appropriate for the publication of this journal.

1. One of the fundamental concerns is the whole process of the annealing appears to be driven by a single modification at one of ~nine bps, where most of them are GC bps. One would expect the annealing process should largely be driven and dominated by GC bp interactions, and thus intuitively would consider an alternative 3-state kinetic model to count for the RD data:

I'd suggest re-analyze the RD and CEST data under this alternative 3-state model. I suspect this 3-state model could also at least be possible, especially for long helices where it's both thermodynamically and kinetically favorable to form, regardless of what happens in one of the positions.

2. Explain why the presence of Mg²⁺ has an effect on RD of m6AMP.

3. Include data for C8, C10 of m6AMP at all temperatures in Suppl. Table 1. Do RD data measured at C2, C10, and C8 agree with each other at various temperatures? From the data presented in the ms, it's not clear.

4. The main conclusion appears to be made based on the 2-state fit/constrained 3-state simulation of the data acquired at T=65. It would be helpful if the authors discuss what would be at 37C, a temperature more relevant to the physiological condition?

5. Reconcile/rewording seemingly contradictory tandem statements "...in which the methylamino group rotates into the energetically favored syn isomer. Although such a conformation is predicted to be highly energetically disfavored..."

6. Given its relatively small size by Mw of the RNAs in this study, S/N would possibly permit detection of the two HB present in the m6Aanti:U basepair. Did the authors attempt to detect the presence of the two hydrogen bonds in dsRNAanti using HNN-cosy for the (A)N1---H-N3(U) H-bond and NOE for

the another?

7. I suggest plotting the CS difference between m6A and m62A and m6A vs. residue number at three temperatures in Ext. fig.7. The plots would better illustrate the chemical shift differences in adjacent residues.

8. The proposed 4-state model is an underdetermined system with multiple unknowns and assumptions. It's always possible to fit kinetic models, given enough number of "floating" parameters to fit based on my experience and literature. I'd suggest the authors discuss the possibility.

9. Fig 5 appears to suggest even distributions of 50% of IF and CS at T=55C in the 4-state model. More convincing experimental data would be direct detection of species, which might be feasible in this case given the equal population distribution using a high-field spectrometer.

Reviewer #3 (Remarks to the Author):

Review of Nature Communications Liu et al. "A quantitative model predicts how m6A reshapes the kinetic landscape of nucleic acid hybridization and conformational transitions"

The manuscript of Liu et al. describes detailed conformational analysis of the m6A methylation modification in RNA. Posttranscriptional m6A methylation and demethylation are critical features of regulation in eukaryotic organisms. While the biological importance of these modifications has been clearly outlined, their structural and biophysical effects on RNA structure have remained unclear; m6A modified As can pair with U similar to unmodified As, but only in the anti-form of the methyl amino group. Multiple groups have shown that m6A destabilizes the ability of RNA-RNA interactions to form, but the origins of this kinetic effect were unknown. Liu et al apply elegant NMR exchange methods to identify the syn-anti conformational exchange of the methyl group as the key modulator of RNA base pairing kinetics. The preferred syn form blocks pairing with U, slowing duplex formation while transition to anti in the final structure leads to no effect on the dissociation rates. This effect occurs by pathways that depend on whether the isomerization of the m6A group occurs before or after base pairing. The authors determine the rates of these processes and create detailed kinetic models that explain the m6A effects on stability and exchange. The results of this rigorous and important study are of broad interest to the readers of Nature Communications and beyond, and finally place the effects of m6A on a firm physical foundation. As such the manuscript absolutely deserves publication in Nature Communications. Below I make some suggestions and possible additional data that will improve the final published manuscript.

1. The presentation of the relaxation dispersion measurements and fitting is not clear in the main text. The methods and mathematics, albeit complex, are presented in the methods and supp material. This reader, who understands a bit of NMR, always needs to be reminded of the exchange theory used to extract the excited states. I assume non-NMR readers may struggle further. A nice figure panel in either Fig 1 or better still Fig 2 that outlines the exchange measurements and makes better sense of the field dependent curves fits in subsequent panels would go miles in helping a reader through the data

2. The duplex association and dissociation rates were extracted from NMR exchange data at high temperature. I would like to see these rates confirmed by an alternative method, such as T-jump or fluorescence or another method independent of NMR. If this is too much work, a temperature dependence of the rates that give the appropriate thermodynamic parameters. In short, it would be nice to "trust" the rates as measured by NMR as true association and dissociation rates. These parameters are critical to the models presented here, and methods are referred to in (21).

3. Relatedly, the authors jump around to make measurements at different temperatures. I think I extracted the logic of moving from RT to 55 or 65C, in order to populate excited states and accelerate rates to make the exchange measurements. However to the novice reader I think this point might be lost. A clearer outline of the experimental logic would help greatly.

4. It would have been interesting to see the effect of m6A on base triple formation, since now the methyl group, even in anti, would inhibit triple formation.
5. All these minor critiques are meant to improve the impact of the paper to the non-NMR reader. These results are exciting and explain for example our own results on tRNA and release factor binding in the A site. Publish away! Jody Puglisi

Reviewer #1 (Remarks to the Author):

This paper describes a detailed relaxation dispersion NMR investigation of the role of m6A methylation in regulating strand formation and melting. Using sophisticated NMR dynamics methods, a series of nucleic acid constructs, different temperatures and labeling that stabilizes conformational preferences, a model is described consisting of separate induced fit and conformational selection pathways that explains the kinetics of duplex formation/melting. Notably, the intermediate formed in the induced fit pathway is one where the syn conformation is retained in the duplex state – the first time that such a conformation has been observed. However, the authors provide convincing evidence that indeed this is in fact the case. This has potential implications for recognition of nucleic acids by cognate proteins and further enhances our knowledge of the different structures that these molecules can form.

We thank the reviewer for his/her positive comments.

The work represents an important contribution to the literature and a stunning example of how to dissect what appears to be a complex 4 state pathway. I have only 1 simple question. Could the authors provide an intuitive explanation for why at 65oC the syn double strand conformation is much lower than at 55oC, that is, what influences the DH of the process so significantly?

Based on RD NMR measurements on the hairpin construct, in which the exchange contribution from the ssRNA state is essentially eliminated (see Fig. 4a-c and Extended Data Fig. 6a), the dsRNA^{syn} population is similar at 55°C (~1.1%) and 65°C (~1.6%). The dsRNA^{syn} species was not directly observable at 65°C in our RD NMR experiments on the duplex construct because its exchange contribution was probably masked by the much larger contribution due to the ssRNA state with population ~22%. By reducing the population of the ssRNA to 5%, we were able to unmask and observe the dsRNA^{syn} intermediate at 55°C in the duplex construct.

The reviewer may be curious about why the flux along the IF pathway is lower at 65°C (10%) versus 55°C (50%). Since the hybridization step is rate-limiting for both CS and IF pathways, the flux is primarily determined by the population of the ssRNA species and annealing rate constants. The smaller flux along the IF pathway at 65°C versus 55°C can be attributed to a slower annealing rate along the IF pathway at 65°C due to a 2-fold reduction in population of the ssRNA^{syn} relative to ssRNA^{anti} and comparatively 2.5-fold slower annealing rate constants for ssRNA^{syn} along the IF pathway relative to ssRNA^{anti} along the CS pathway.

To clarify these points, we added the following two paragraphs to the main text on pages 12 and 20, respectively:

“Although we did not observe any evidence for the IF dsRNA^{syn} intermediate, simulations indicate that its RD contribution was probably masked by the larger RD contribution from the ssRNA with population ~22%. We therefore repeated the CEST measurements at a slightly lower temperature T = 55°C. This reduced the ssRNA population to ~5%, but it remained large enough to permit accurate measurements of hybridization kinetics using NMR RD.

Repeating the measurements at a different temperature also allowed us to test the robustness of the CS model.”

“The smaller flux along the IF pathway at 65°C versus 55°C can be attributed to a slower annealing rate along the IF pathway at 65°C due to a 2-fold reduction in population of the ssRNA^{syn} relative to ssRNA^{anti} and comparatively 2.5-fold slower annealing rate constant of ssRNA^{syn} along the IF pathway relative to ssRNA^{anti} along the CS pathway.”

Reviewer #2 (Remarks to the Author):

Liu et al. present a study of a quantitative model to predict kinetic behaviors of nucleic acid consisting of m⁶A methylation in hybridization and conformational transitions. The m⁶A methylation is an important posttranscriptional modification in the regulation of gene expression. This study shows that m⁶A modulates the annealing/melting behavior of nucleic acids through syn/anti-conversion and conformational selection pathways. The study is highly significant; the approach and data analysis appear to be rigorous. After addressing the following concerns, the manuscript is appropriate for the publication of this journal.

We thank the reviewer for his/her positive comments.

1. One of the fundamental concerns is the whole process of the annealing appears to be driven by a single modification at one of ~nine bps, where most of them are GC bps. One would expect the annealing process should largely be driven and dominated by GC bp interactions, and thus intuitively would consider an alternative 3-state kinetic model to count for the RD data:

ssRNA-synm⁶A + ssRNA-complementary <-> dsRNA-with-syn "bulge"m⁶A <-> dsRNA

I'd suggest re-analyze the RD and CEST data under this alternative 3-state model. I suspect this 3-state model could also at least be possible, especially for long helices where it's both thermodynamically and kinetically favorable to form, regardless of what happens in one of the positions.

We thank the reviewer for this comment - we entirely agree. Indeed, in the original manuscript, the model suggested by the reviewer was one of the main pathways that we analyzed, referring to it as the "induced-fit (IF)" pathway (see updated Fig. 1c). In this pathway, m⁶A ssRNA^{syn} anneals with the complementary strand to form an intermediate in which all bps with the exception of the m⁶A(syn) residue are formed. We structurally characterized this intermediate and found that m⁶A(syn) pairs with its partner U via a single H-bond. This is then followed by isomerization of the methylamino group to form the duplex product. The other pathway which we focused on early in the paper was the "conformational selection" (CS) pathway in which isomerization occurs prior to annealing (Fig. 2a). Indeed, we found that annealing proceeds with comparable flux along the two pathways, although the relative flux varied slightly depending on temperature, sequence, and buffer conditions.

In the original submission, we noted that the CS pathway alone fails to account for the CEST data measured at 55°C (Fig. 3). While we did not mention this explicitly in the original submission, the IF pathway alone also fails to account for the data measured at 65°C (Extended Data Fig. 3c) and 55°C (Extended Data Fig. 4e). In the revision, we have included Extended Data Fig. 3c and Extended Data Fig. 4e showing that the IF pathway alone cannot effectively model the CEST data.

New Extended Data Fig. 3c. Constrained 3-state fits to the IF pathway for dsGGACU^{m6A} C2/C8 CEST at 65°C.

New Extended Data Fig. 4e. Constrained 3-state fits to the IF pathway for dsGGACU^{m6A} C2/C8 CEST at 55°C.

We also added a method section entitled “3-state IF simulations and constrained fits for the dsGGACU^{m6A} RD data measured at $T = 65^\circ C$ and $55^\circ C$ ” on page 45, describing the approach we used to perform IF pathway constrained fitting.

In addition, to avoid confusion, we changed the introduction on page 4 and updated Fig. 1c to acknowledge early on the possibility for the induced fit pathway in addition to conformational selection:

“Kinetic mechanisms involving binding and conformational change can occur via pathways wherein the conformational change occurs prior or post binding¹. We therefore hypothesized that m^6A could slow hybridization via at least two pathways in which isomerization of the methylamino group occurs either before or following duplex formation (Fig. 1c). In the conformational selection (CS) pathway, hybridization proceeds via an unpaired intermediate (ssRNA^{anti}) with m^6A in the energetically disfavored anti conformation (Fig. 1c). In the induced fit (IF) pathway, the more populated ssRNA^{syn} species with m^6A in the syn

conformation initially hybridizes to form a double-stranded intermediate (dsRNA^{syn}) that entails the loss of at least one Watson-Crick H-bond between m⁶A and the partner uridine (Fig 1a). This is then followed by isomerization to form the Watson-Crick bp (dsRNA^{anti}) with m⁶A in the anti conformation (Fig. 1c). To date, there has been no evidence for the dsRNA^{syn} intermediate.”

2. Explain why the presence of Mg²⁺ has an effect on RD of m⁶AMP.

Based on the m⁶AMP RD data measured at 25°C (Supplementary Table 1), Mg²⁺ slightly increased the population of the m⁶A(*anti*) excited state from 5.8±0.7% to 9.4±0.8%, while k_{ex} was within error (2241±62 s⁻¹ vs. 2357±46 s⁻¹). Consistent with this finding, we have previously shown that m⁶A is not as destabilizing when introduced into duplexes (by ~0.2 kcal/mol) in the presence versus absence of Mg²⁺. We speculate that these small effects could be due to Mg²⁺ binding to the N7 position of m⁶A² although this requires further verification.

We have added the following statement on page 10.

“Consistent with this interpretation, RD measurements on the m⁶A monomer reveal that 3 mM Mg²⁺ stabilizes the anti relative to syn isomer by ~0.5 kcal/mol³, and correspondingly, the destabilizing effects of m⁶A on RNA duplexes is reduced by ~0.2 kcal/mol in the presence of 3 mM Mg²⁺ relative to the absence of Mg²⁺ (Supplementary Table 1).”

3. Include data for C8, C10 of m⁶AMP at all temperatures in Suppl. Table 1. Do RD data measured at C2, C10, and C8 agree with each other at various temperatures? From the data presented in the ms, it's not clear.

We thank the reviewer for this question. In the case of m⁶AMP, we only measured RD for C2 and C8. The RD profiles for C8 were all flat as was the case for ssGGACU^{m⁶A} (Extended Data Fig. 2a) and the hpGGACU^{m⁶A} (Extended Data Fig. 6a). This is most likely because the C8 chemical shift is insensitive to isomerization of the methylamino group. We did not have C10 labelled m⁶AMP and did not measure C10 RD at natural abundance because the signal was too weak. We did however measure RD for C2, C8 and C10 in ssGGACU^{m⁶A} at 25°C and 37°C and the data is in good agreement, the population and k_{ex} is close to within error (Supplementary Table 1). Similarly, the C2 and C10 CEST data measured at 55°C and 65°C for hpGGACU^{m⁶A} could be combined in a global fit (Supplementary Table 1). Note that C8 is sensitive to duplex melting, which is why we observe C8 RD at 55°C and 65°C in dsRNA constructs (Extended Data Fig. 4a, 5) but not at 37°C (Extended Data Fig. 5) due to the low population (< 0.1%) of the ssRNA species.

In the revision, we updated Supplementary Table 1 to include included the C8 RD parameters in which only R_1 and R_2 can be reliably fitted due to lack of any detectable exchange.

4. The main conclusion appears to be made based on the 2-state fit/constrained 3-state simulation of the data acquired at T=65. It would be helpful if the authors discuss what would be at 37C, a temperature more relevant to the physiological condition?

We thank the reviewer for raising this important point. In the original submission, we briefly addressed this point, but this was buried in the Methods section on page 51 “Predict m^6A -induced slowdown of DNA hybridization in the mouse genome”. In the revision, we moved this discussion from the methods to discussion on page 27:

“Our NMR measurements had to be performed under high temperature conditions so that hybridization falls within the detection limits of RD. However, we were able to observe isomerization of the methylamino group in both ssRNA and dsRNA at $T = 37^\circ C$ (Extended Data Fig. 2b and 6a). Based on the temperature dependence of the hybridization steps in the CS and IF pathways, our model predicts (see Methods) that m^6A will slow down annealing by ~ 5 -fold while minimally impacting the melting rate consistent with our measurements at higher temperatures. A comparable level of slowdown in annealing is also obtained when predicting the m^6A induced slowdown at $T = 37^\circ C$ using rate constants for hybridization of unmethylated RNA reported previously⁴ at $T = 37^\circ C$ and assuming that m^6A destabilizes dsRNA by 1 kcal/mol ⁵ (see Methods).”

5. Reconcile/rewording seemingly contradictory tandem statements “...in which the methylamino group rotates into the energetically favored *syn* isomer. Although such a conformation is predicted to be highly energetically disfavored...”

We thank the reviewer for pointing this out. We meant to say the *syn* conformation is energetically favored in unpaired m^6A while it is predicted to be highly disfavored in dsRNA. We revised this sentence on page 14 as follows:

*“Although never observed previously, one possibility is that the new intermediate is a dsRNA conformation in which the methylamino group rotates into the *syn* conformation. Such a conformation is predicted to be highly energetically disfavored in dsRNA, given the loss of at least one Watson-Crick H-bond. However, this loss in energetic stability would be partly compensated for by a gain in stability of $\sim 1.5 \text{ kcal/mol}$ from restoring the energetically favored *syn* isomer.”*

6. Given its relatively small size by Mw of the RNAs in this study, S/N would possibly permit detection of the two HB present in the $m^6A_{anti}:U$ basepair. Did the authors attempt to detect the presence of the two hydrogen bonds in dsRNA_{anti} using HNN-cosy for the (A)N1---H-N3(U) H-bond and NOE for the another?

The conformation of the $m^6A_{anti}:U$ base pair in the dsGGACU ^{m^6A} has been extensively characterized previously based on NOEs in a prior study³. Additionally, prior structural studies using NMR⁶ and crystallography⁷ have shown that $m^6A_{anti}:U$ forms the two Watson-Crick hydrogen bonds. We made changes to page 4 to clarify this point:

*“Rather, when paired with uridine, the methylamino group rotates into the energetically disfavored *anti* isomer and forms a canonical m^6A-U Watson-Crick bp that retains both (A)N1...H-N3(U) and (A)N6...H-O4(U) H-bonds (Fig. 1a).”*

7. I suggest plotting the CS difference between m^6A and m^6A vs. residue number

at three temperatures in Ext. fig.7. The plots would better illustrate the chemical shift differences in adjacent residues.

We thank the reviewer for this suggestion. In the revision, we added plots showing the chemical shift differences in Extended Data Fig. 7c. We observe upfield shifted C8, C2 and C1' in the dimethylated residue in both RNA duplexes, consistent with DFT calculations (Extended Data Fig. 7c). In addition, the chemical shifts of the dimethylated A partner and neighboring residues also show perturbations, indicating that dimethylation only locally affects RNA structure.

New Extended Data 7c. Chemical shift perturbations comparing dimethylated and m⁶A modified dsGGACU and dsA6 RNA constructs (left). Shown on the right are the chemical shift perturbations for C2, C8 and C1' measured for dsGGACU and dsA6 and calculated using DFT.

8. *The proposed 4-state model is an underdetermined system with multiple unknowns and assumptions. It's always possible to fit kinetic models, given enough number of "floating" parameters to fit based on my experience and literature. I'd suggest the authors discuss the possibility.*

We thank the reviewer for this thoughtful suggestion. We agree that with a 4-state model there is a risk of over-fitting the data. It is for this reason that we initially assessed the CS+IF 4-state model using explicit simulations in which no parameter was allowed to float and in which approximate rate constants for each step were measured independently using appropriate constructs or conditions tailored to isolate as much as possible a particular step. These values and the 4-state model were then used to simulate the CEST profiles without any adjustable parameter (see Extended Data Fig. 9a). We observed good agreement between simulated and experimentally measured CEST data. Afterwards, we performed a constrained 4-state fit allowing the parameters to float around these average values by an amount determined by experimental uncertainty (one standard deviation). To clarify these points, we added the following paragraph to the methods section on page 43:

"When dealing with 3-state or 4-state exchange, there is always a danger of overfitting the RD data. For this reason, we initially performed simulations in which all of the relevant kinetic rate constants, populations, $\Delta\omega$, R_1 and R_2 of the different species were approximated to

values measured experimentally using the appropriate RNA constructs (Extended Data Fig. 2b, 6a and Supplementary Table 1). These values were then used in a 3-state or a 4-state simulation to simulate CEST profiles without any adjustable parameters. We then performed constrained fits in which the parameters (population, rate constants, $\Delta\omega$, R_1 and R_2) were allowed to float by an amount determined by the experimentally measured uncertainty (one standard deviation)."

9. Fig 5 appears to suggest even distributions of 50% of IF and CS at T=55C in the 4-state model. More convincing experimental data would be direct detection of species, which might be feasible in this case given the equal population distribution using a high-field spectrometer.

While the reviewer is correct that the flux through the CS and IF pathways is 50%, the two intermediates have low populations and/or are in fast exchange on the NMR timescale, as a consequence they cannot be observed directly as separate resonances. This underscores a general point, which is that many reaction intermediates are low-populated and short-lived, and difficult to resolve using many conventional biophysical approaches. The low-populated species would be difficult to directly observe in NMR spectra even with higher field due to fast exchange. We do observe separate resonances for the starting duplex reactant and single stranded product, because here the exchange is slow on the NMR timescale.

Reviewer #3 (Remarks to the Author):

Review of Nature Communications Liu et al. "A quantitative model predicts how m6A reshapes the kinetic landscape of nucleic acid hybridization and conformational transitions"

The manuscript of Liu et al. describes detailed conformational analysis of the m6A methylation modification in RNA. Posttranscriptional m6A methylation and demethylation are critical features of regulation in eukaryotic organisms. While the biological importance of these modifications has been clearly outlined, their structural and biophysical effects on RNA structure have remained unclear; m6A modified As can pair with U similar to unmodified As, but only in the anti-form of the methyl amino group. Multiple groups have shown that m6A destabilizes the ability of RNA-RNA interactions to form, but the origins of this kinetic effect were unknown. Liu et al apply elegant NMR exchange methods to identify the syn-anti conformational exchange of the methyl group as the key modulator of RNA base pairing kinetics. The preferred syn form blocks pairing with U, slowing duplex formation while transition to anti in the final structure leads to no effect on the dissociation rates. This effect occurs by pathways that depend on whether the isomerization of the m6A group occurs before or after base pairing. The authors determine the rates of these processes and create detailed kinetic models that explain the m6A effects on stability and exchange. The results of this rigorous and important study are of broad interest to the readers of Nature Communications and beyond, and finally place the effects of m6A on a firm physical foundation. As such the manuscript absolutely deserves publication in Nature Communications. Below I make some suggestions and possible additional data that will improve the final published manuscript.

We thank Jody for his positive comments.

1. The presentation of the relaxation dispersion measurements and fitting is not clear in the main text. The methods and mathematics, albeit complex, are presented in the methods and supp material. This reader, who understands a bit of NMR, always needs to be reminded of the exchange theory used to extract the excited states. I assume non-NMR readers may struggle further. A nice figure panel in either Fig 1 or better still Fig 2 that outlines the exchange measurements and makes better sense of the field dependent curves fits in subsequent panels would go miles in helping a reader through the data

These are great suggestions. We have added the full CS+IF pathway schematic in Fig. 1c, laying out all the definitions of the various species and kinetic rate constants.

Revised Fig. 1. **c**. Schematic of the CS+IF model with kinetic rate constants defined as follows: k_1 and k_{-1} are the forward and backward rate constants for methylamino isomerization in ssRNA, respectively. k_2 and k_{-2} are the forward and backward rate constants for methylamino isomerization in dsRNA, respectively. $k_{on,anti}$ and $k_{off,anti}$ are the annealing and melting rate constants, respectively when m⁶A adopts *anti* conformation in both ssRNA and dsRNA. $k_{on,syn}$ and $k_{off,syn}$ are the annealing and melting rate constants, respectively when m⁶A adopts *syn* conformation in both ssRNA and dsRNA.

We also added a brief introduction to the $R_{1\rho}$ and CEST experiment on page 7:

“NMR RD experiments can be used to characterize conformational exchange between a dominant ground state (GS) and short-lived low-populated “excited-state” (ES). The $R_{1\rho}$ experiment measures the line-broadening contribution (R_{ex}) to the transverse relaxation rate (R_2) during a relaxation period in which a continuous radiofrequency (RF) field is applied with variable power (ω_{SL}) and frequency (ω_{RF}). The RF field reduces the R_{ex} contribution in a manner dependent on ω_{SL} and ω_{RF} and the exchange parameters of interest (see below). The RD profiles are typically displayed by plotting the measured $R_2 + R_{ex}$ as a function of ω_{SL} and ω_{RF} . For detectable exchange, a peak is observed centered at the difference between the chemical shift of the GS and ES ($-\Delta\omega$, assuming $\omega_{GS} = 0$ and $\omega_{ES} = \Delta\omega$). The CEST experiment measures the impact of conformational exchange on longitudinal GS magnetization during a relaxation period following application of a continuous RF field with variable power (ω_{SL}) and frequency (ω_{RF}). When applied on resonance with the ES, the RF field saturates the ES magnetization and this saturation can be transferred via conformational exchange to the GS. This typically results in a reduced signal intensity for the GS and a minor dip centered at $\omega_{ES} = \Delta\omega$ when the RF is on resonance with ES. A major dip is also observed centered at $\omega_{GS} = 0$ when the RF field is on resonance with the GS. The dependencies of $R_2 + R_{ex}$ ($R_{1\rho}$) or the GS signal intensity (CEST) on ω_{SL} and ω_{RF} can be fit to the Bloch-McConnell equations describing n -site exchange to determine exchange parameters of interest (see below).”

We also provide a description of the $R_{1\rho}$ and CEST data when they are first introduced on page 8:

“Shown in Fig. 2c on the left is the CEST profile recorded for the m^6 A-C10 methyl carbon in ssGGACU m6A as a function of RF. As is typical for CEST profiles, a major dip is observed when the RF field is on-resonance with the GS chemical shift at $\Delta\omega = 0$. In addition, a minor dip was observed indicative of conformational exchange with a sparsely populated ES. The dip was observed at a chemical shift $\Delta\omega_{C10} = \omega_{ES} - \omega_{GS} = 3$ ppm, which was in good agreement with the value predicted for the anti isomer ($\Delta\omega_{C10} = 3$ -5 ppm) using density functional theory (DFT) calculations⁸ (Methods). Shown in Fig. 2c on the right is the $R_{1\rho}$ profile measured for m^6 A-C2 in ssGGACU m6A as a function of RF field. A peak was observed at $-\Delta\omega_{C2} = 0.6$ ppm indicative of conformational exchange. A similar C2 RD was observed in methylated but not unmethylated AMP, as expected if the RD is reporting on isomerization (Extended Data Fig. 2a).”

2. The duplex association and dissociation rates were extracted from NMR exchange data at high temperature. I would like to see these rates confirmed by an alternative method, such as T-jump or fluorescence or another method independent of NMR. If this is too much work, a temperature dependence of the rates that give the appropriate thermodynamic parameters. In short, it would be nice to “trust” the rates as measured by NMR as true association and dissociation rates. These parameters are critical to the models presented here, and methods are referred to in (21).

We thank Jody for these suggestions. We did perform something analogous to the requested comparison when we first introduced the NMR RD methodology in the prior (ref 21) paper⁹. In this prior ref 21, we benchmarked the NMR RD method using two 12-mer DNA duplexes. The kinetic rate constants for duplex hybridization measured by NMR RD fell within the expected range of values reported previously using fluorescence spectroscopy for duplexes of similar length. For example, for 12-mer DNA duplexes, k_{on} ranged between $2 \times 10^5 - 4 \times 10^6 \text{ M}^{-1}\text{s}^{-1}$ compared to our value of $\sim 1.2 \times 10^6 \text{ M}^{-1}\text{s}^{-1}$ while k_{off} ranged between $0.4 - 200 \text{ s}^{-1}$, compared to our value of $5 - 60 \text{ s}^{-1}$. In addition, the kinetic rate constants for duplex hybridization measured by NMR RD exhibited the well-established strong dependence of k_{off} on sequence while k_{on} showed virtually no sequence dependence, in agreement with prior studies¹⁰. In addition, the chemical shifts deduced by NMR RD for both DNA and RNA duplexes were shown to be in excellent agreement with those measured for the isolated single-stranded species, and this internal consistency further supports that the NMR RD data is reporting on the duplex hybridization kinetics.

To further test the NMR RD approach, we followed Jody’s suggestion and performed temperature dependent ^{13}C CEST measurements on dsGGACU m6A . These new data are included in the revision in Extended Data Fig. 11. k_{off} was strongly dependent on temperature as reported in prior studies of duplex hybridization kinetics employing fluorescent spectroscopic techniques^{11,12} and could be fit to a standard van’t Hoff equation ($R^2 = 0.97$). On the other hand, k_{on} showed a much weaker dependence on temperature and exhibited deviations from the Arrhenius behavior ($R^2 = 0.68$), in very good agreement with

prior studies^{11,12}, which also report a weak temperature dependence and non-Arrhenius behavior for k_{on} . For comparison, both the forward and backward rate constants for methylamino isomerization showed a strong temperature dependence and both could be fit to a standard van't Hoff equation ($R^2 > 0.96$, see Extended Data Fig. 2c, 6d).

Following Jody's suggestion, we used the NMR RD data to determine thermodynamic parameters $\Delta H_{\text{anneal}}^\circ$ and $\Delta S_{\text{anneal}}^\circ$ for duplex annealing based on the temperature dependence of p_{B} (ssRNA population) which was used to deduce the value of $\Delta G_{\text{anneal}}^\circ$ at each temperature. We then compared these thermodynamic parameters to counterparts measured previously for dsGGACU^{m6A} using UV melting experiments³. Indeed, we find good agreement between the two measurements particularly for ΔG° ; the difference is < 0.2 kcal/mol (Extended Data Fig. 11b). Although the $\Delta H_{\text{anneal}}^\circ$ and $\Delta S_{\text{anneal}}^\circ$ values are not within error, small deviations are to be expected given that these parameters are generally not as well determined as for $\Delta G_{\text{anneal}}^\circ$ and that different techniques were used employing very different duplex concentrations (NMR ~ 1 mM versus UV ~ 3 μM).

In addition, we also observed good agreement between $\Delta G_{\text{anneal}}^\circ$ measured using CEST and UV melting experiments for 9 additional DNA/RNA duplexes at temperatures varying between 45°C to 65°C (Extended Data Fig. 11c). We suspect that the small systematic differences probably arise in part due to small differences in the temperature calibration on the two instruments, given that the $\Delta G_{\text{anneal}}^\circ$ values are reported near T_{m} making them particularly sensitive to temperature. In addition, duplex-duplex interactions at the higher duplex concentration used in the NMR experiments could explain the slightly higher duplex stability observed by NMR relative to UV.

Extended Data Fig. 11. **Independent tests of NMR RD measurements of m^6A RNA hybridization.** **a**, Temperature dependence of the melting (k_{off}) and annealing (k_{on}) rate constants of $dsGGACU^{m6A}$. **b**, Temperature dependence of annealing free energy ΔG_{anneal}° , which is derived from population of $ssGGACU^{m6A}$ measured by CEST. **c**, Comparison of thermodynamic parameters measured using optical melting experiments (UV)³ and temperature dependent CEST experiments (CEST) for $dsGGACU^{m6A}$. **d**, Comparison of ΔG_{anneal}° measured⁹ for 10 DNA/RNA duplexes at temperatures ranging from 45°C to 65°C.

In the revision, we make sure to point the reader to the prior paper describing the original RD NMR approach on page 3:

“Recently, we developed and validated an NMR relaxation-dispersion (RD)¹³⁻¹⁵ based method to measure the hybridization kinetics in DNA and RNA duplexes⁹. Using this approach, we showed that m^6A preferentially slows the apparent rate of RNA duplex annealing by ~5-10-fold while having little effect on the apparent rate of duplex melting⁹ (Fig. 1b).”

We also included a description of the new temperature dependent hybridization kinetics in the methods section on page 37 along with the new Extended Data Fig. 11:

“Validation of NMR RD measurements on m^6A RNA hybridization. We have previously⁹ shown that hybridization kinetics measured from NMR RD on unmodified DNA and RNA duplexes are consistent with those measured using other techniques employing fluorescence spectroscopy. As an additional test, we performed temperature dependent RD measurements for $dsGGACU^{m6A}$ (Extended Data Fig. 11a). The annealing rate constant k_{on}

did not have a strong temperature dependence, consistent with prior studies reporting non-Arrhenius behavior for k_{on} in unmodified duplexes^{11,12}. On the other hand, the melting rate constant k_{off} showed a strong temperature dependence, also consistent with prior studies^{11,12}. The extrapolated annealing thermodynamic parameters including $\Delta G_{anneal}^{\circ}$, $\Delta H_{anneal}^{\circ}$ and $\Delta S_{anneal}^{\circ}$ measured from NMR experiments are in good agreement with those measured from UV melting experiments³ (Extended Data Fig. 11b-c). We also observed a good agreement between the annealing free energy ($\Delta G_{anneal}^{\circ}$) measured using CEST and UV melting experiments for 9 additional DNA/RNA duplexes at temperatures ranging from 45°C to 65°C⁹ (Extended Data Fig. 11d).”

3. Relatedly, the authors jump around to make measurements at different temperatures. I think I extracted the logic of moving from RT to 55 or 65C, in order to populate excited states and accelerate rates to make the exchange measurements. However to the novice reader I think this point might be lost. A clearer outline of the experimental logic would help greatly.

We agree. The reason we started at high temperatures is because the hybridization kinetics can be better characterized using NMR RD owing to larger ssDNA population and faster exchange kinetics. To clarify this point, we added the following sentence on page 10:

“The CEST experiments were performed at high temperature because at 37°C, the ssRNA is too lowly populated (<0.1%) and the hybridization is too slow (<50 s⁻¹) to be effectively characterized by RD.”

4. It would have been interesting to see the effect of m6A on base triple formation, since now the methyl group, even in anti, would inhibit triple formation.

We thank Jody for sharing this interesting idea! We agree that m⁶A should in principle disrupt the base triple structure. We do plan to survey for base triple structures and examine the effect of m⁶A.

5. All these minor critiques are meant to improve the impact of the paper to the non-NMR reader. These results are exciting and explain for example our own results on tRNA and release factor binding in the A site. Publish away! Jody Puglisi

Thank you Jody!

References

- 1 Hammes, G. G., Chang, Y. C. & Oas, T. G. Conformational selection or induced fit: a flux description of reaction mechanism. *Proc Natl Acad Sci U S A* **106**, 13737-13741, doi:10.1073/pnas.0907195106 (2009).
- 2 Zheng, H., Shabalin, I. G., Handing, K. B., Bujnicki, J. M. & Minor, W. Magnesium-binding architectures in RNA crystal structures: validation, binding preferences, classification and motif detection. *Nucleic Acids Res* **43**, 3789-3801, doi:10.1093/nar/gkv225 (2015).
- 3 Liu, B. *et al.* A potentially abundant junctional RNA motif stabilized by m(6)A and Mg(2). *Nat Commun* **9**, 2761, doi:10.1038/s41467-018-05243-z (2018).
- 4 Cisse, II, Kim, H. & Ha, T. A rule of seven in Watson-Crick base-pairing of mismatched sequences. *Nat Struct Mol Biol* **19**, 623-627, doi:10.1038/nsmb.2294 (2012).
- 5 Kierzek, E. & Kierzek, R. The thermodynamic stability of RNA duplexes and hairpins containing N6-alkyladenosines and 2-methylthio-N6-alkyladenosines. *Nucleic Acids Res* **31**, 4472-4480 (2003).
- 6 Roost, C. *et al.* Structure and thermodynamics of N6-methyladenosine in RNA: a spring-loaded base modification. *J Am Chem Soc* **137**, 2107-2115, doi:10.1021/ja513080v (2015).
- 7 Huang, L., Ashraf, S., Wang, J. & Lilley, D. M. Control of box C/D snoRNP assembly by N6-methylation of adenine. *EMBO Rep* **18**, 1631-1645, doi:10.15252/embr.201743967 (2017).
- 8 Kimsey, I. J., Petzold, K., Sathyamoorthy, B., Stein, Z. W. & Al-Hashimi, H. M. Visualizing transient Watson-Crick-like mispairs in DNA and RNA duplexes. *Nature* **519**, 315-320, doi:10.1038/nature14227 (2015).
- 9 Shi, H. *et al.* NMR Chemical Exchange Measurements Reveal That N(6)-Methyladenosine Slows RNA Annealing. *J Am Chem Soc* **141**, 19988-19993, doi:10.1021/jacs.9b10939 (2019).
- 10 Wyer, J. A., Kristensen, M. B., Jones, N. C., Hoffmann, S. V. & Nielsen, S. B. Kinetics of DNA duplex formation: A-tracts versus AT-tracts. *Phys Chem Chem Phys* **16**, 18827-18839, doi:10.1039/c4cp02252a (2014).
- 11 Sorgenfrei, S. *et al.* Label-free single-molecule detection of DNA-hybridization kinetics with a carbon nanotube field-effect transistor. *Nat Nanotechnol* **6**, 126-132, doi:10.1038/nnano.2010.275 (2011).
- 12 Wallace, M. I., Ying, L., Balasubramanian, S. & Klenerman, D. Non-Arrhenius kinetics for the loop closure of a DNA hairpin. *Proc Natl Acad Sci U S A* **98**, 5584-5589, doi:10.1073/pnas.101523498 (2001).
- 13 Rangadurai, A., Szymaski, E. S., Kimsey, I. J., Shi, H. & Al-Hashimi, H. M. Characterizing micro-to-millisecond chemical exchange in nucleic acids using off-resonance R1rho relaxation dispersion. *Prog Nucl Magn Reson Spectrosc* **112-113**, 55-102, doi:10.1016/j.pnmrs.2019.05.002 (2019).
- 14 Palmer, A. G., 3rd & Massi, F. Characterization of the dynamics of biomacromolecules using rotating-frame spin relaxation NMR spectroscopy. *Chem Rev* **106**, 1700-1719, doi:10.1021/cr0404287 (2006).
- 15 Palmer, A. G., 3rd. Chemical exchange in biomacromolecules: past, present, and future. *J Magn Reson* **241**, 3-17, doi:10.1016/j.jmr.2014.01.008 (2014).

REVIEWERS' COMMENTS

Reviewer #1 (Remarks to the Author):

The authors have addressed all of my concerns and this excellent manuscript is ready for publication.

Reviewer #2 (Remarks to the Author):

The authors have made their best efforts to address the reviewers' questions and concerns. In particular, the authors clearly understand the co-existence of the IF and CS pathways and present them appropriately in the current revision. As of what they presented in the original submission, it could be misconstrued as if the CS is the main pathway driving the annealing of the two otherwise totally complementary strands. On the biological relevance of their finding: the co-existence of the two pathways could only be significant at an elevated temperature, such as near T_m , at which base-stacking and H-bonding interactions, due to the near-strand separation at equilibrium, are no longer the overwhelmingly driving forces. At the physiologically relevant temperature such as 37 degrees, one would expect the IF to be the dominant pathway, and the second step in IF is much more efficient than the 1st step of the CS because of a much higher local "concentration" after the formation of the nearly complete duplex, and the IF pathway is negligible at least, or irrelevant. The authors do not have direct quantitative experimental data at 37 degrees. Thus, I suggest the authors revise the statement in the abstract by adding quantifier "Using NMR relaxation dispersion recorded at elevated temperatures ... with a singly hydrogen-bonded low-populated (1%) mismatch-like conformation in which the methylamino group is syn" to avoid misconception.

I'm otherwise satisfied with other revisions.

Reviewer #3 (Remarks to the Author):

The authors have very nicely addressed my comments and greatly improved the presentation of their data. The manuscript is now ready for publication.

point-by-point response to the reviewers' comments

Reviewer #1 (Remarks to the Author):

The authors have addressed all of my concerns and this excellent manuscript is ready for publication.

We thank this reviewer for his/her positive comments.

Reviewer #2 (Remarks to the Author):

The authors have made their best efforts to address the reviewers' questions and concerns. In particular, the authors clearly understand the co-existence of the IF and CS pathways and present them appropriately in the current revision. As of what they presented in the original submission, it could be misconstrued as if the CS is the main pathway driving the annealing of the two otherwise totally complementary strands. On the biological relevance of their finding: the co-existence of the two pathways could only be significant at an elevated temperature, such as near T_m , at which base-stacking and H-bonding interactions, due to the near-strand separation at equilibrium, are no longer the overwhelmingly driving forces. At the physiologically relevant temperature such as 37 degrees, one would expect the IF to be the dominant pathway, and the second step in IF is much more efficient than the 1st step of the CS because of a much higher local "concentration" after the formation of the nearly complete duplex, and the IF pathway is negligible at least, or irrelevant. The authors do not have direct quantitative experimental data at 37 degrees. Thus, I suggest the authors revise the statement in the abstract by adding quantifier "Using NMR relaxation dispersion recorded at elevated temperatures ... with a singly hydrogen-bonded low-populated (1%) mismatch-like conformation in which the methylamino group is syn" to avoid misconception. I'm otherwise satisfied with other revisions.

We thank this reviewer for his/her positive comments.

In the revised abstract, we now explicitly point out the temperatures used in the NMR experiments and also point out that the mechanism is established quantitatively at elevated temperatures as suggested by the reviewer.

ABSTRACT: N^6 -methyladenosine (m^6A) is a post-transcriptional modification that controls gene expression by recruiting proteins to RNA sites. The modification also slows biochemical processes through mechanisms that are not understood. Using temperature-dependent (20°C – 65°C) NMR relaxation dispersion, we show that m^6A pairs with uridine with the methylamino group in the *anti* conformation to form a Watson-Crick base pair that transiently exchanges on

the millisecond timescale with a singly hydrogen-bonded low-populated (1%) mismatch-like conformation in which the methylamino group is *syn*. This ability to rapidly interchange between Watson-Crick or mismatch-like forms, combined with different *syn:anti* isomer preferences when paired (~1:100) versus unpaired (~10:1), explains how m⁶A robustly slows duplex annealing without affecting melting at elevated temperatures via two pathways in which isomerization occurs before or after duplex annealing. Our model quantitatively predicts how m⁶A reshapes the kinetic landscape of nucleic acid hybridization and conformational transitions, and provides an explanation for why the modification robustly slows diverse cellular processes.

Reviewer #3 (Remarks to the Author):

The authors have very nicely addressed my comments and greatly improved the presentation of their data. The manuscript is now ready for publication.

We thank Jody for his positive comments.